EMBO
reports

# A cell atlas of adult muscle precursors uncovers early events in fibre-type divergence in *Drosophila*

Maria Paula Zappia[1], Lucia de Castro[1], Majd M Ariss[1] (iD), Holly Jefferson[1] (iD), Abul BMMK Islam[2] (iD) &
Maxim V Frolov[1,*] (iD)

## Abstract

In *Drosophila*, the wing disc-associated muscle precursor cells give rise to the fibrillar indirect flight muscles (IFM) and the tubular direct flight muscles (DFM). To understand early transcriptional events underlying this muscle diversification, we performed single-cell RNA-sequencing experiments and built a cell atlas of myoblasts associated with third instar larval wing disc. Our analysis identified distinct transcriptional signatures for IFM and DFM myoblasts that underlie the molecular basis of their divergence. The atlas further revealed various states of differentiation of myoblasts, thus illustrating previously unappreciated spatial and temporal heterogeneity among them. We identified and validated novel markers for both IFM and DFM myoblasts at various states of differentiation by immunofluorescence and genetic cell-tracing experiments. Finally, we performed a systematic genetic screen using a panel of markers from the reference cell atlas as an entry point and found a novel gene, *Amalgam* which is functionally important in muscle development. Our work provides a framework for leveraging scRNA-seq for gene discovery and details a strategy that can be applied to other scRNA-seq datasets.

**Keywords** differentiation; muscle; single cell; transcriptome
**Subject Categories** Development; Methods & Resources

## Introduction

Muscle fibres exhibit significant variability in biochemical, mechanical and metabolic properties, which are defined by the needs and specialized functions of each muscle. The *Drosophila* adult skeletal muscle represents an ideal system to dissect the transcriptional events regulating muscle diversity. In the adult fly, the thoracic muscle contains two types of flight muscles, the indirect flight muscles (IFM) and the direct flight muscles (DFM), that have distinct structure, positioning, patterning and specialized function

(Lawrence, 1982). The IFM are fibrillar muscles that provide power to fly, whereas the DFM are tubular muscles required for proper wing positioning. Fibre fate is specified by the transcriptional factors *extradenticle* (*exd*), *homothorax* (*hth*) and *spalt major* (*salm*), which control the expression of fibre-specific structural genes and sarcomeric components during myofibrillogenesis at early pupal stages (Schönbauer *et al*, 2011; Bryantsev *et al*, 2012). However, there is very little information about the extent of divergence of the transcription programmes in the muscle precursor cells that give rise to these two muscle types.

The muscle precursor cells that form both the IFM and DFM, generally referred to as myoblasts, are associated with the wing imaginal discs. This pool of myoblasts located in the adepithelial layer contains the adult muscle precursor (AMP) cells, which are specified early in development (Bate *et al*, 1991; Dobi *et al*, 2015). The AMPs are considered muscle-committed transient stem cells and share some features with the vertebrate adult muscle stem cells called satellite cells (Figeac *et al*, 2007). During larval stages, the AMPs are activated and undergo extensive proliferation, first symmetrically and then asymmetrically, in which the AMPs self-renew and generate a differentiating myoblast. By the late third instar larval stage, myoblasts reach a population size of around 2,500 cells (Gunage *et al*, 2014). This large pool of myoblasts then fuse and form the adult striated muscles (Gunage *et al*, 2017). Interestingly, the cells that will form the IFM are located on the presumptive notum and show expression of both *vestigial* (*vg*) and *cut* (*ct*), whereas the cells that will give rise to the DFM are located near the presumptive wing hinge and only show very high levels of expression of *ct* but no expression of *vg* (Sudarsan *et al*, 2001; Fig 1A). It has been suggested that such divergence is maintained by both intrinsic and extrinsic signals, the latter emanating from epithelial cells of the wing disc. One such signal is Wingless (Wg), secreted from the notum, that maintains *vg* expression in IFM myoblasts and establishes a boundary between IFM and DFM myoblasts (Sudarsan *et al*, 2001). However, with the exception of *vg*-specific expression in IFM myoblasts and *ct* differential expression, no other genes are known to distinguish these two groups of cells, raising the question of what other changes in gene expression are also taking place. Compounding the issue is the lack of knowledge about the level of heterogeneity within each group of cells. Yet, this is important for

1   Department of Biochemistry and Molecular Genetics, University of Illinois at Chicago, Chicago, IL, USA
2   Department of Genetic Engineering and Biotechnology, University of Dhaka, Dhaka, Bangladesh
    *Corresponding author. E-mail: mfrolov@uic.edu

the interpretation of experiments in which transplantation of the labelled wing disc-associated myoblast cells into larval hosts led to an indiscriminate contribution to the developing adult muscles. It was suggested that the specification of myoblasts at the larval stage is not yet definite, and therefore, myoblasts can still adapt to changing environmental cues (Lawrence & Brower, 1982). Whether this conclusion is applicable to an entire pool of myoblasts or to a more naïve population of myoblasts that is uniquely capable of such transformation is unknown.

To investigate the early divergence in the transcriptional programmes between DFM and IFM myoblasts, we performed single-cell RNA-seq experiments of the proximal third instar wing disc and constructed a high-resolution reference cell atlas comprising 4,544 myoblast cells, which yields 1.8× cellular coverage. We found that IFM and DFM myoblasts have distinct transcriptional signatures indicating that the genetic regulatory networks driving each muscle type diverge prior to fibre fate specification. Unexpectedly, the atlas revealed that IFM and DFM myoblasts are highly heterogeneous and each group contains distinct populations representing cells at various states of differentiation. Finally, by combining the scRNA-seq approach with an RNAi based genetic screen and genetic cell-tracing experiments, we identified new genes that are important for skeletal muscle development.

## Results

### A single-cell atlas of the proximal wing imaginal disc identifies diverse cell types

We performed scRNA-seq to identify the differences in the transcriptional profiles of two subtypes of myoblasts that give rise to direct and indirect flight muscles. Wild-type wandering third instar larvae were collected between 110 and 135 h after egg laying (AEL) at 25°C. Wing discs were dissected, cut along the presumptive hinge to remove most of the wing pouch and enrich the sample for myoblasts that are located in the adepithelial layer of the notum (Fig 1A). Dissected tissue was dissociated into single-cell suspension and processed using the Drop-seq protocol (Macosko et al, 2015). Single-cell transcriptomes of eight independent replicates of

two wild-type stocks *1151-GAL4* and *1151 > mCherry-RNAi* were sequenced, and data were processed using an integrative analysis in the Seurat 3 package (Stuart et al, 2019). After filtering poor quality cells, the Uniform Manifold Approximation and Projection (UMAP) dimensionality reduction algorithm was used to visualize cell populations (Fig 1B, see Materials and Methods for more details). To eliminate batch effects, clusters of cells that were not evenly represented among the replicates were removed (Fig EV1A and B, Appendix Fig S1). Using these stringent criteria, we retained 11,527 high-quality cells to generate a reference wild-type atlas of the proximal wing imaginal disc with an average of 500 genes per cell (Fig EV1C–E).

Unsupervised graph-based clustering identified 26 cell clusters (Fig 1C), each exhibiting a distinct gene expression signature (Fig 1D, Datasets EV1 and EV2). Nineteen clusters represented epithelial cells based upon the expression of the epithelial marker *Fasciclin 3* (*Fas3*) (Bate & Martinez Arias, 1991; Fig 1E). The remaining seven clusters comprised 4,544 cells that lacked *Fas3* expression but showed high levels of expression of the myoblast-specific genes *Zn finger homeodomain 1* (*zfh1*) and *Holes in muscle* (*Him*) (Lai et al, 1991; Soler & Taylor, 2009), indicating that these are myoblasts. Accordingly, the principal component analysis revealed that the majority of gene expression variance among the cells was accounted by epithelial- and muscle-specific genes (PC1, Fig EV1F). Using anti-Fas3 and anti-Zfh1 antibodies, epithelial cells and myoblasts were visualized in the wing discs by immunofluorescence (Fig 1F).

To further explore the differences between myoblast and epithelial clusters, we examined the list of differentially expressed genes between these two cell types (Dataset EV3). First, the expression of *ct* and *vg* was examined in the dataset (Fig 1G). These genes were found in myoblasts as well as in some clusters of epithelial and tracheal cells. As previously reported, the sensory organ precursor cells (SOPs, cluster ES_1), and the tracheoblasts of the spiracular branches (Trachea_2, white arrowhead Fig 1H) showed high levels of *ct* expression (Blochlinger et al, 1990; Pitsouli & Perrimon, 2010). The epithelial clusters of the dorso-ventral boundary (Epi_9 and Epi_10) showed *vg* expression (Williams et al, 1991). In addition to the canonical muscle-specific markers *twi*, *Mef2*, *Him* and *zfh1*, other muscle-related genes, such as *Secreted protein, acidic,*

---

**Figure 1. Single-cell atlas of the proximal wing imaginal disc identifies diverse cell types.**

A   Origin of the flight muscles. Left panel: Trachea, air sac primordium, IFM myoblasts and DFM myoblasts in the adepithelial layer overlaying the wing disc epithelium. Right panel: Lateral view of the adult flight muscles in the thorax.

B   Workflow for droplet-based scRNA-seq. Wandering third instar larval wing discs were dissected, and pouch was removed and dissociate into single-cell suspension. Droplets containing unique barcoded beads and single cells were collected. Following library preparation, sequencing data were aligned, and a gene-cell expression matrix was generated and analysed using Seurat for identification of variable genes and unsupervised cell clustering based on gene expression similarity.

C   Annotated cell type, including 6,711 epithelial, 272 tracheal and 4,544 myoblast cells, in UMAP plot of the reference single-cell atlas.

D   RNA expression heatmap showing the top differentially expressed gene markers for each cluster of the reference single-cell atlas dataset. Cells (column) are clustered by the expression of the main marker genes (row).

E   Average expression level of the genes *Fas3* (left panel) and *zfh1* (right panel) used as markers to assign epithelial and myoblast cells, respectively, in the reference dataset.

F   Confocal single plane image of third instar larval wing disc and orthogonal views of the disc stained with anti-Zfh1 (red) and anti-Fas3 (green).

G   Dot plot showing the expression levels of the marker genes identified for the myoblast, epithelial and tracheal cells across the 26 clusters of the reference cell atlas. Colour intensity represents the average normalized expression level. Dot diameter represents the fraction of cells expressing each gene in each cluster. Gene expression for each cell was normalized by the total expression, and then, the expression of each gene was scaled.

H   Confocal single plane image of third instar larval wing disc and orthogonal view of *SPARC>GFP* (green) stained with anti-Ct (red) and 4′,6-diamidino-2-phenylindole (DAPI, blue). Full genotype *y-, w-/w-; UAS-GFP/+; SPARC-GAL4/+*.

Data information: Scale bars: 50 μm. White arrowhead: Ct expression in tracheoblasts.

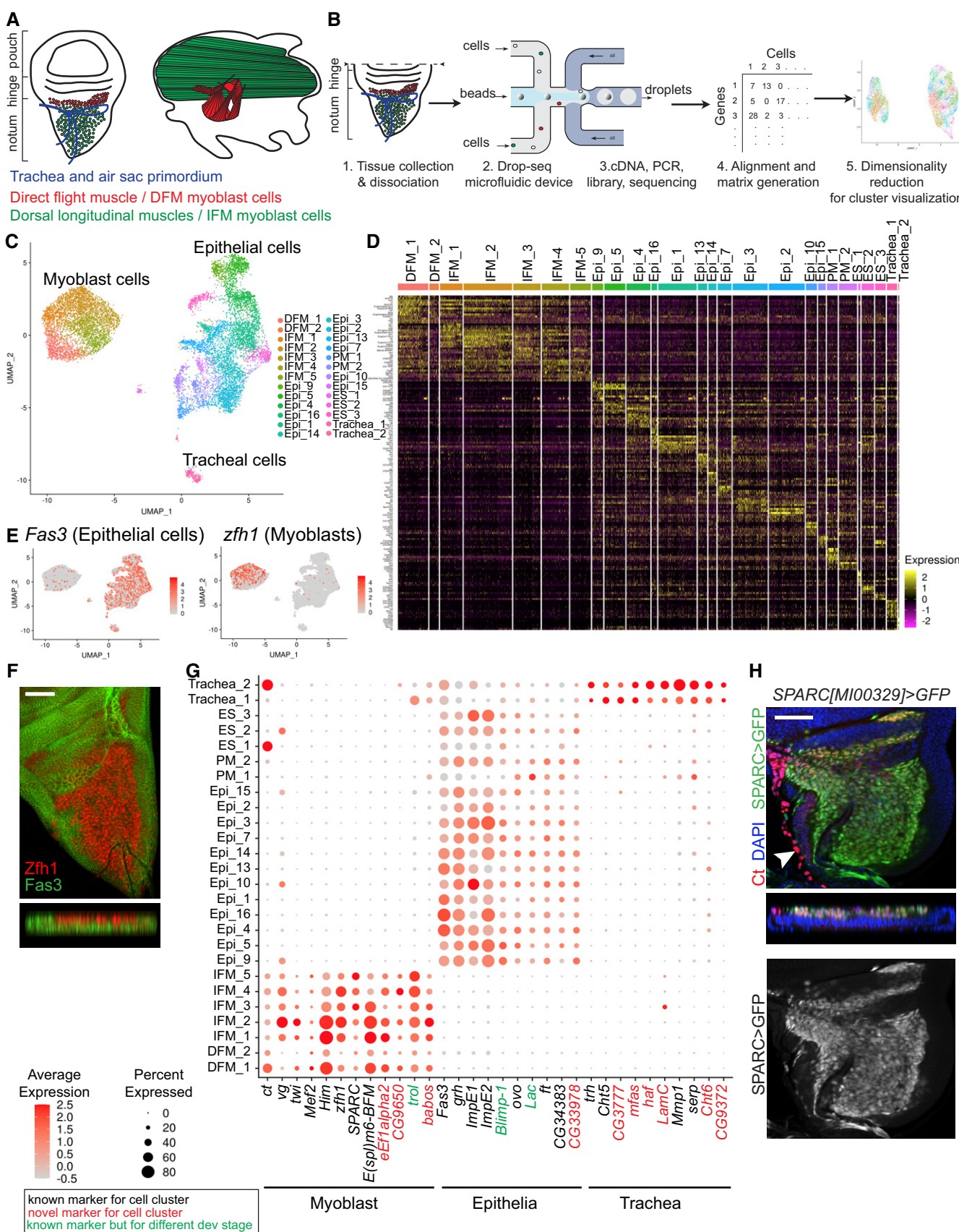

**Figure 1.**

*cysteine-rich* (*SPARC*), *CG9650*, *eukaryotic translation elongation factor 1 alpha 2* (*eEf1alpha2*), *terribly reduced optic lobes* (*trol*) and *babos,* were found to be highly expressed in the myoblast clusters (Fig 1G). Among them, Trol was shown to be expressed in muscle attachment sites in embryonic muscle (Friedrich *et al*, 2000) and *SPARC* in the myoblasts at the larval third instar (Butler *et al*, 2003). We confirmed this result using the *SPARC > GFP* reporter line (Fig 1H).

The assignment of the epithelial clusters was further supported by the specific expression of the *ovo*, *grainy head* (*grh*), *Ecdysone-inducible gene E1* (*ImpE1*), *four-jointed* (*ft*), *Ecdysone-inducible gene E2* (*ImpE2*) and *CG34383* (*Kramer*) genes (Fig 1G), which were reported to be expressed in diverse epithelium, and *Blimp-1* and *Lachesin* (*Lac*) in trachea epithelium (Llimargas *et al*, 2004; Ng *et al*, 2006). We also confirmed the expression of *grh* in the epithelial layer of the wing disc directly using the *grh::GFP* reporter line and showed mutual exclusivity of *grh::GFP* expression with the myoblast marker Twist (Fig EV1G).

Two epithelial clusters (Trachea_1 and Trachea_2) expressed a canonical tracheal marker *trachealess* (*trh*) (Sato & Kornberg, 2002) indicating that these clusters contain tracheal cells. Accordingly, we detected the expression of several known tracheal-specific genes including *Chitinase 5* (*Cht5*), *serpentine* (*serp*) and *Matrix metalloproteinase 1* (*Mmp1*). Interestingly, *pebbled* (*peb*) and *breathless* (*btl*) were expressed exclusively in cells of Trachea_1, while *Ultrabithorax* (*Ubx*) (Brower, 1987), *Gasp*, *Cuticular protein 49Ag* (*Cpr49Ag*), *Pherokine 3* (*Phk-3*), *Cuticular protein 12A* (*Cpr12A*) and *Cystatin-like* (*Cys*) exhibited Trachea_2-specific expression (Fig EV1H). We also identified novel markers for each cluster (Figs 1G and EV1H, Dataset EV4). Trachea_1 likely represents cells from the air sac primordium since one of the marker was *btl* (Sato & Kornberg, 2002). Trachea_2 seems to contain cells from the spiracular branches because it showed *ct* expression (Fig 1G; Pitsouli & Perrimon, 2010). We confirmed this by staining with anti-Ct antibody (white arrowhead, Fig 1H).

We concluded that the single-cell reference atlas contains epithelial cells of the wing disc, myoblasts and tracheal cells and, therefore, accurately represents the cellular diversity of the proximal wing imaginal disc.

## Cells in the epithelial clusters map to spatially distinct regions of the wing disc

Unbiased clustering analysis grouped the epithelial cells of the wing disc into 17 transcriptionally distinct cell clusters and two types of tracheal cells based on their expression profiles. Since the wing discs show a restrictive pattern of gene expression, we wonder whether the single-cell clusters reflect this spatially distinct domains. We selected the marker genes for each cluster and then searched the literature for published *in situ* expression patterns of these genes in the wing disc. The positions of the cell clusters were then mapped to the presumptive adult structures using the cell fate map of the wing disc (Bryant, 1975; Fig 2A). In this way, we assigned the identities of twelve clusters to the disc proper, two clusters to the peripodial membrane and three clusters to cells associated with external sensory organs (Fig 2B and C). The corresponding markers used for assignment as well as the new markers are shown in the feature maps (Figs 2D and EV2A, Dataset EV2) and in the dot plot (Fig 2E). Additionally, the expression pattern of both known and novel markers was confirmed by immunofluorescence (Fig 2F). Below, we detail how cells in each cluster were mapped back to the spatial positions based on the integration of expression signatures of known genes.

Cells in clusters Epi_9 and Epi_5 corresponded to cells spatially localized to the wing blade and the periphery of the wing blade, respectively. This is based on the expression of *nubbin* (*nub*, Fig 2F) (Ng *et al*, 1995), *rotund* (*rn*) (St Pierre *et al*, 2002), *ventral veins lacking* (*vvl*) (de Celis *et al*, 1995) and *CG17278* (Mohit *et al*, 2006) in both clusters. Moreover, *wg*, which is found in the inner ring of the periphery of the wing blade (Fig 2F; Couso *et al*, 1993; Terriente *et al*, 2008), was only expressed in Epi_5. Accordingly, *CG30069* and *Lipid storage droplet-2* (*Lsd-2*), which are expressed in wing blade (Butler *et al*, 2003; Fauny *et al*, 2005), were Epi_9-specific markers, while *Zn finger homeodomain 2* (*zfh2*), which is found at the periphery of the wing blade (Whitworth & Russell, 2003), was an Epi_5-specific marker.

Cells that belonged to both Epi_4 and Epi_16 mapped to the wing hinge, with Epi_4 cells located distal to Epi_16. This is due to the expression of *Sox box protein 15* (*Sox15*), the top marker for Epi_4 that is restricted to the hinge (Crémazy *et al*, 2001) between the inner and outer rings (Dichtel-Danjoy *et al*, 2009), along with *zfh2* and *apterous* (*ap*) (Cohen *et al*, 1992). In contrast, cells of Epi_16 expressed high levels of *dachshund* (*dac*) (Mardon *et al*, 1994), *crossveinless 2* (*cv-2*) (Conley *et al*, 2000), *Daughters against dpp* (*Dad*, Fig 2F; Tsuneizumi *et al*, 1997), *salm/spalt-related* (*salr*) (de Celis *et al*, 1996a) and *bifid* (*bi*) (Sun *et al*, 1995). The expression pattern of these genes mostly overlaps in the central area of the hinge along the A/P boundary close to the presumptive lateral notum. Accordingly, a subset of cells of this cluster showed low levels of expression of *araucan* (*ara*) (Gómez-Skarmeta *et al*, 1996) and *hairy* (*h*) (Usui *et al*, 2008), which are localized to the lateral notum. Since *decapentaplegic* (*dpp*, Fig 2F) was also expressed in Epi_16 and is a marker of the anterior cells in the A/P boundary (Posakony *et al*, 1991), we reasoned that the cells of cluster Epi_16

**Figure 2. Cells in epithelial clusters map to spatially distinct regions of the wing disc.**

A  Schematic representation of the wing imaginal disc and its relationship to the adult wing. The cell fate map of the wing disc is adapted from (Bryant, 1975).
B  Subset of 6,983 epithelial cells in UMAP plot of the reference single-cell atlas.
C  Approximate map of the epithelial clusters identified in the reference single-cell atlas dataset to distinct position over the disc proper.
D  Average expression level of the genes used as known markers to assign each epithelial cluster in the reference atlas dataset. Cells coloured by the expression of *nub*, *wg*, *zfh2*, *Sox15*, *grn*, *bnl*, *eyg*, *Ance*, *Idg4* and *drm*.
E  Dot plot showing the expression levels of the top marker genes identified for the epithelial cells across 17 clusters of the reference cell atlas dataset.
F  Confocal single plane image of third instar larval wing disc stained with anti-Nub, anti-Wg, anti-Svp, anti-Hairy (red), reporting expression of *dpp-lacZ*, *Dad-lacZ* (anti-β-gal, red) and counterstained with DAPI (cyan). Scale bars: 100 μm. Full genotypes: *P{CaryP}attP2* (top and bottom panels), *P{PZ}dpp[10638]; +*, and *y- w-; +; P{lacW} Dad[j1E4]*.

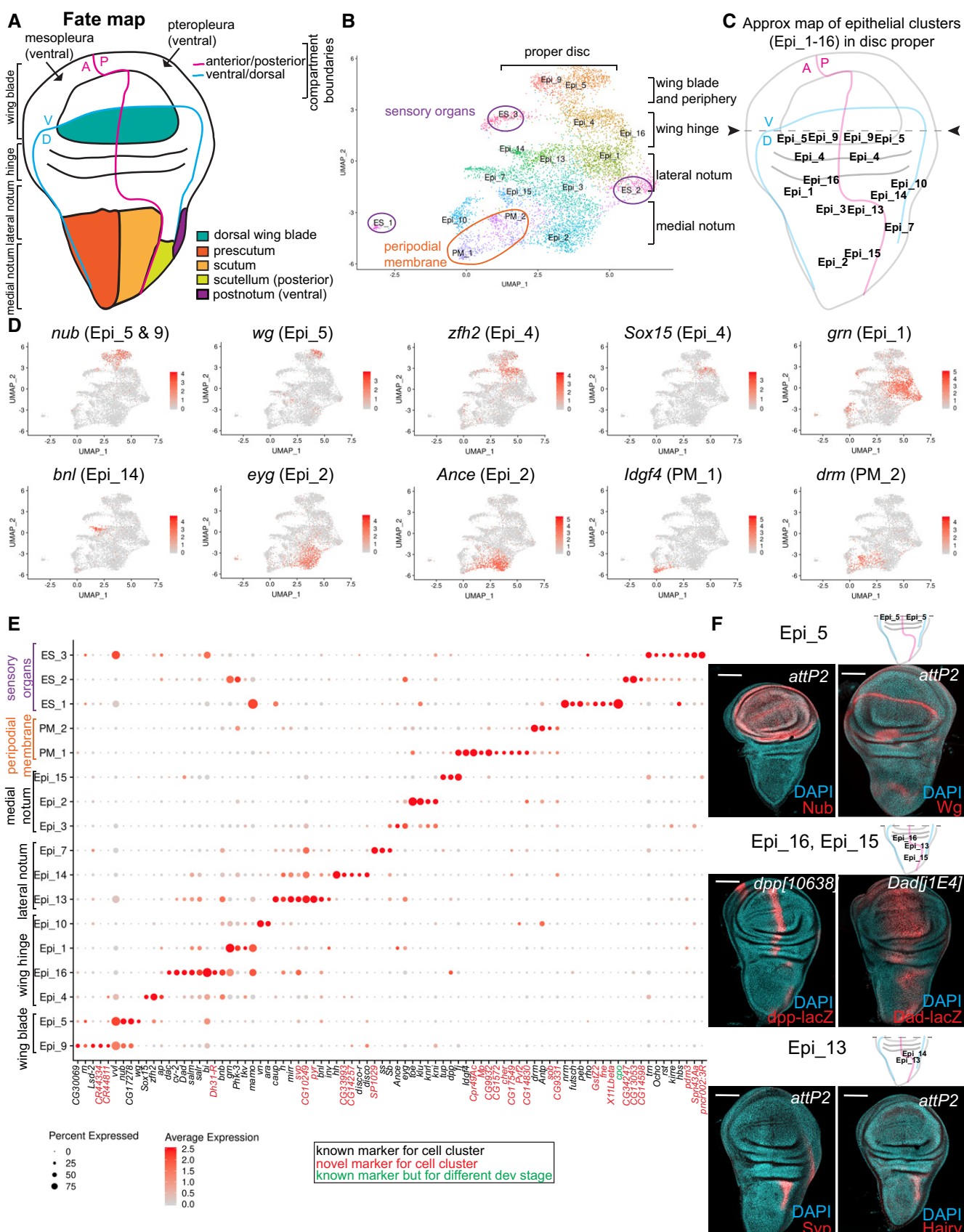

**Figure 2.**

are mostly positioned in the anterior compartment near the A/P boundary of the hinge.

Epi_1 cells were likely localized in the anterior hinge close to the presumptive lateral notum. These cells showed high levels of *grain* (*grn*), which is expressed in the hinge (Brown & Castelli-Gair Hombría, 2000), *thickveins* (*tkv*), which is found in the most anterior and posterior area of presumptive notum (Brummel *et al*, 1994), and *Phk-3*, which is expressed in the anterior hinge and presumptive notum (Klebes *et al*, 2005). Epi_10 cells only expressed high levels of two genes, *knirps* (*kni*) and *knirps-like* (*knrl*), which are found in the posterior area of the hinge (Lunde *et al*, 1998).

Cells that clustered into Epi_13 and Epi_14 were localized along the lateral heminotum near the hinge as they share common markers, such as *ara*, *caupolican* (*caup*) (Gómez-Skarmeta *et al*, 1996) and *mirror* (*mirr*) (Kehl *et al*, 1998). Epi_13 showed specific expression of *vein* (*vn*) (Simcox *et al*, 1996) and high levels of *h* (Fig 2F; Usui *et al*, 2008), whereas Epi_14 showed specific expression of the posterior markers *invected* (*inv*) (Coleman *et al*, 1987) and *hedgehog* (*hh*) (Tabata & Kornberg, 1994), and *branchless* (*bnl*), which straddles the A/P compartment border near the scutum and hinge (Sato & Kornberg, 2002). Thus, the approximate location of Epi_13 cells were near the presumptive scutum in the lateral heminotum (near scutellum) and Epi_14 cells were likely in the posterior lateral heminotum (near scutum). The expression pattern of *seven up* (*svp*), a novel marker for cluster Epi_13, was examined by immunofluorescence using anti-Svp antibodies. As expected, the localization of Svp matched with the expression pattern of Hairy in Epi_13 cells, thus validating the predicted map of cell clusters over the disc (Fig 2F). The approximate location of Epi_7 cells was near the post-notum in the posterior heminotum, since the expression of Epi_7 marker *disco-r* is restricted to the ventral edge of the wing (Grubbs *et al*, 2013) and the expression of another Epi_7 marker *disco* is restricted to the region giving rise to the post-alar bristles near the presumptive lateral scutum (Cohen *et al*, 1991).

Known markers for the anterior presumptive notum, including *eyegone* (*eyg*) (Aldaz *et al*, 2003), *twin of eyg* (*toe*) (Yao *et al*, 2008) and *klumpfuss* (*klu*) (Klein & Campos-Ortega, 1997) were highly expressed in Epi_2 and Epi_3. Because *spineless (ss)* (Duncan *et al*, 1998) and *Stubble* (*Sb*) (Ibrahim *et al*, 2013) were detected at higher levels in Epi_3, whereas *Angiotensin-converting enzyme* (*Ance*) (Siviter *et al*, 2002) was specifically in Epi_2, we inferred that cells contributing to Epi_3 were positioned in the anterior lateral notum near the hinge and to Epi_2 were in the anterior medial notum. The markers for Epi_15 cells were *four-jointed* (*fj*) and *tailup* (*tup*), which localize to the prospective notum (Cho & Irvine, 2004; de Navascues & Modolell, 2007). Because *dpp* (Fig 2F), which was also expressed in Epi_15, downregulates the Iro-C genes (*ara*, *caup* and *mirr*) in the medial notum (Cavodeassi *et al*, 2002), we reasoned that cells of this cluster are most likely positioned anterior to the A/P boundary in the medial notum.

Two cell clusters, PM_1 and PM_2, corresponded to the peripodial membrane, a layer of squamous cells overlaying the epithelial cells of the disc proper. PM_1 was likely mapping to the dorsal peripodial epithelium, as it expressed both *Imaginal disc growth factor 4* (*Idgf4*) (Butler *et al*, 2003) and *Ance*. Two additional peripodial markers, *drumstick* (*drm*) (Benitez *et al*, 2009) and *Antennapedia* (*Antp*) (Levine *et al*, 1983), were highly expressed in PM_2.

The expression of *sca* in ES_1, ES_2 and ES_3 indicated that these clusters represented either developing external sensory organs or neighbouring cells contributing to their development (Fig EV2B). The expression of *sca* was highest in ES_1, thus indicating that this cluster represents the sensory organ precursor cells (SOPs) that are undergoing cytodifferentiation into neuronal receptors. Concordantly, SOPs markers *neuromusculin* (*nrm*) (Ghysen & O'Kane, 1989), *futsch* (Klein & Campos-Ortega, 1997), *peb* (Giraldez *et al*, 2002) and *rhomboid* (*rho*) (Sturtevant *et al*, 1993) were highly expressed in cells of ES_1 (Fig 2E). A common marker for ES_2 and ES_3 was *tartan* (*trn*), which is expressed in macrochaete SOP but not in microchaete SOP of the anterior wing margin (Chang *et al*, 1993), and *Ocho*, which is found in the external sensory organs (Lai *et al*, 2000). Thus, ES_2 and ES_3 most likely represent bristle precursors cells. However, ES_3 expressed *bifid* (*bi*) and *vvl*, and ES_2 showed high levels of *grn*, *Phk-3* and *Sb*. Therefore, ES_3 cells were likely surrounding SOPs in the presumptive hinge and wing blade, whereas ES_2 cells are located in the anterior hinge or anterior presumptive lateral notum region. Concordantly, we found *roughest* (*rst*), *kin of irre* (*kirre*) and *hibris* (*hbs*) in both ES_2 and ES_3. These genes encodes components of the heterophilic Irre Cell Recognition Module associated with cells surrounding SOPs in the anterior wing margin (Linneweber *et al*, 2015).

Thus, we conclude that unbiased cell clustering of our epithelial cell data performed in Seurat identifies spatially distinct regions of the wing disc. However, for most clusters, the top markers were expressed in only about 50% of cells in cluster, which is likely due to drop-outs, a common issue in scRNA-seq (Kharchenko *et al*, 2014).

## The single-cell transcriptome atlas reveals a large compendium of adult muscle precursors

Next, we examined the transcriptional profile of adepithelial myoblast cells in greater detail. The wing disc-associated adult myoblasts give rise to two distinct sets of flight muscles, IFM and DFM. However, our unbiased cell clustering revealed seven distinct myoblast clusters (Fig 3A). We assigned IFM and DFM myoblasts based on the differential expression of *ct* and the expression of *vg* (Fig 3B), the only two known markers for DFM and IFM myoblasts (Sudarsan *et al*, 2001). Notably, five clusters showed a high *vg* and low *ct* pattern of expression, indicating that these represent IFM myoblasts (IFM_1-5), while two clusters showed no *vg* and high levels of expression of *ct*, and therefore, were classified as DFM myoblasts (DFM_1-2, Fig 3B). Seurat analysis further revealed that the level of *zfh1* expression is higher in IFM myoblasts than in DFM myoblasts (Fig 3B). Since *zfh1* is considered a general marker for the entire pool of myoblasts (Lai *et al*, 1991), we carefully examined Zfh1 expression by co-staining the wing disc with anti-Zfh1 antibody and anti-Ct antibody. Indeed, Zfh1 and Ct were differentially expressed between the IFM and DFM myoblasts, thus confirming the scRNA-seq results (Fig 3C). The expression of the flight muscle determinant genes, such as *salm*, *exd* and *hth*, was either too low or unchanged among the myoblast clusters, suggesting they may be induced later in development to determine distinct muscle types (Schönbauer *et al*, 2011). However, it is still possible that low gene detection could be due to a technical limitation of Drop-seq.

    

To further examine the differences in the transcriptional programmes between DFM and IFM myoblasts, the expression of the marker genes for each type of myoblast was visualized across cell populations (Figs 3B and EV3A, and Dataset EV5). There was a relatively large panel of specific markers for DFM myoblasts. By contrast, only *vg*, *zfh1*, *nop5* (Vorbrüggen *et al*, 2000) and *naked cuticle* (*nkd*), which is involved in embryonic muscle patterning (Volk & VijayRaghavan, 1994), were highly expressed in IFM myoblast clusters (Fig 3B). Among the DFM myoblast markers, we found *kirre* (Fig 3D), a marker of muscle founder cells in DFM myoblasts (Kozopas & Nusse, 2002), which is consistent with DFM *de novo* formation. We also found *lateral muscles scarcer* (*lms*) expression restricted to DFM precursors, as previously reported (Muller *et al*, 2010). Other DFM-specific markers are genes whose roles were described in other types of muscles, including *midline* (*mid*) (Kumar *et al*, 2015), *Anaplastic lymphoma kinase* (*Alk*) (Englund *et al*, 2003; Lee *et al*, 2003) and *wing blister* (*wb*) (Martin *et al*, 1999), which are involved in embryonic myogenesis, and *sprout* (*sty*), which regulates maturation of adult founder cells in the abdomen (Dutta *et al*, 2005). Finally, the panel of DFM markers included *arginine kinase (Argk)*, *Neurotactin* (*Nrt*), *Amalgam* (*Ama*) and *Tenascin accessory* (*Ten-a*), whose function in muscle has not been investigated yet. We confirmed cell-type-specific expression of five of these novel markers using *Ama-GAL4* enhancer trap, GFP-tagged reporters *Argk::GFP*, *Ten-a::GFP, nkd::GFP* and anti-Nrt antibody (Fig 3E–I). In most cases, the wing discs were counterstained with both anti-Zfh1 and anti-Ct antibodies to visualize IFM and DFM myoblasts. We further showed the co-expression of Ama with Nrt, and Argk with Nrt in DFM cells, thus confirming the specificity of their expression in DFM myoblasts (Fig EV3B).

While examining the pattern of *Ama-GAL4* driver in the wing disc, we noticed that *Ama* expression appears to extend into some IFM myoblasts in the anterior distal part of the prescutum (yellow arrowhead, Fig 3I). We confirmed this result by *fluorescent in situ hybridization* (FISH) that showed *Ama* expression in DFM myoblasts and also in a small subset of IFM myoblasts (asterisk and yellow arrowhead, respectively, Fig 3J). To unequivocally determine which muscles Ama-expressing cells contribute to, we performed genetic cell-tracing experiments using G-TRACE (Evans *et al*, 2009). In this technique, GFP marks the cells that expressed *Ama* in the past (*GAL4* lineage), while RFP shows the *Ama* expression at the time of the analysis (active GAL4). In the third instar larval wing disc, both GFP and RFP expression were localized primarily to the region of DFM myoblasts, and in very few IFM myoblasts (asterisk and yellow arrowhead, respectively, Fig 3K). The pattern of *Ama-GAL4* expression matched the pattern of *Ama in situ* hybridization, thus validating *Ama-GAL4* as an accurate reporter of *Ama* expression (Fig 3K). Interestingly, GFP was also detected in some IFM myoblasts, thus supporting that *Ama* was indeed expressed in a small subset of IFM myoblasts earlier in development (Figs 3K and EV3C). Accordingly, we detected RFP expression in the developing DFM at 28 h APF by the wing hinge region, but not in the mature DFM of adults, where only GFP was detected (Figs EV3D and 3L). Additionally, only GFP signal was present in mature IFM of adults (Fig EV3E). We conclude that *Ama* is expressed in DFM myoblasts and in a small subset of IFM myoblasts at larval stages. This is consistent with the results from scRNA-seq

showing *Ama* expression in the DFM clusters, along with a small percentage of cells in cluster IFM_1 (Fig 3B).

We conclude that DFM and IFM myoblasts are transcriptionally distinct and, in addition to the known markers *ct* and *vg*, these cells express a panel of muscle-specific markers during larval development. However, the expression of several DFM-specific markers can be detected in some cells in the IFM myoblast cluster IFM_1, albeit at a lower level.

## Exploring the heterogeneity of IFM and DFM precursors

Next, we set out to determine the basis of cell heterogeneity that drives clustering of DFM and IFM myoblasts into seven clusters. The expression of the marker genes for each cell population was visualized using a dot plot (Fig 4A, and Dataset EV6). We began by examining the expression of *twist* (*twi*) and *Him* that are responsible for keeping myoblasts in an undifferentiated state and whose downregulation is required for differentiation (Anant *et al*, 1998; Soler & Taylor, 2009). Both *twi* and *Him* were highly expressed in two IFM clusters, IFM_1 and IFM_2, indicating that these clusters comprise undifferentiated myoblasts. By contrast, the clusters IFM_3, IFM_4 and IFM_5 likely contained differentiating myoblasts as *twi* and *Him* were downregulated in these cells. The expression of *twi* and *Him* is controlled by Notch (Anant *et al*, 1998; Rebeiz *et al*, 2002; Bernard *et al*, 2006; Soler & Taylor, 2009). Concordantly, numerous *E(Spl)* genes that are known targets of the Notch pathway and commonly used to assess Notch activity (Zacharioudaki & Bray, 2014) were expressed in IFM_1 and IFM_2 (Figs 4A and EV4A). Interestingly, although the *E(spl)* genes were still expressed in IFM_3, the levels of expression of both *twi* and *Him* were low, indicating that this cluster may represent myoblasts undergoing transition from the undifferentiated state (IFM_1 and IFM_2) to a more differentiated state (IFM_4 and IFM_5). Similarly, we conclude that DFM_1 represents undifferentiated DFM myoblasts because of the high levels of expression of *twi*, *Him* and *E(spl)* genes, and the differentiating myoblasts are grouped into the DFM_2 cluster, which has low *twi*, *Him* and *E(spl)* gene expression. Intriguingly, we found specific expression of *E(spl)mdelta-HLH* in DFM_1, but not in IFM_1-3 (Fig 4A), suggesting that the Notch-dependent downstream response in DFM myoblasts may differ from IFM myoblasts. Additionally, we detected the expression of both *Actin 57B* (*Act57B*) and *Actin 87B* (*Act87E*) in the DFM_1 cluster, which is in agreement with previous findings (Butler *et al*, 2003). Finally, we examined the expression of known markers for muscle differentiation, including genes encoding proteins required for myoblast fusion (*sns*, *robo*, *sli*, *blow*, *mbc*, *Vrp1*, *WASp*, *Arp2*, *rst*, *hbs*, *sing*, *Ced-12*, *Hem*), myotube attachment (*kon*, *drl*, *if*, *mys*, *Ilk*, *rhea*), sarcomerogenesis (*Act88F*, *Act79B*), and myogenic regulators (*Mef2*, *ewg*), in IFM_4 and IFM_5. However, the RNA levels of these genes were too low to further delineate the differentiated state of these clusters (Dataset EV1).

The spatial expression of *E(spl)* genes in third larval wing disc has been extensively studied (Singson *et al*, 1994; de Celis *et al*, 1996b; Wurmbach *et al*, 1999; Lai *et al*, 2000). Interestingly, *E(spl) m6-BFM*, one of the top markers of IFM_1-3 clusters (Fig 4A), was expressed in the specific region of the adepithelial layer near the anterior region of the presumptive lateral heminotum (Fig 4B; Wurmbach *et al*, 1999; Lai *et al*, 2000; Aradhya *et al*, 2015).

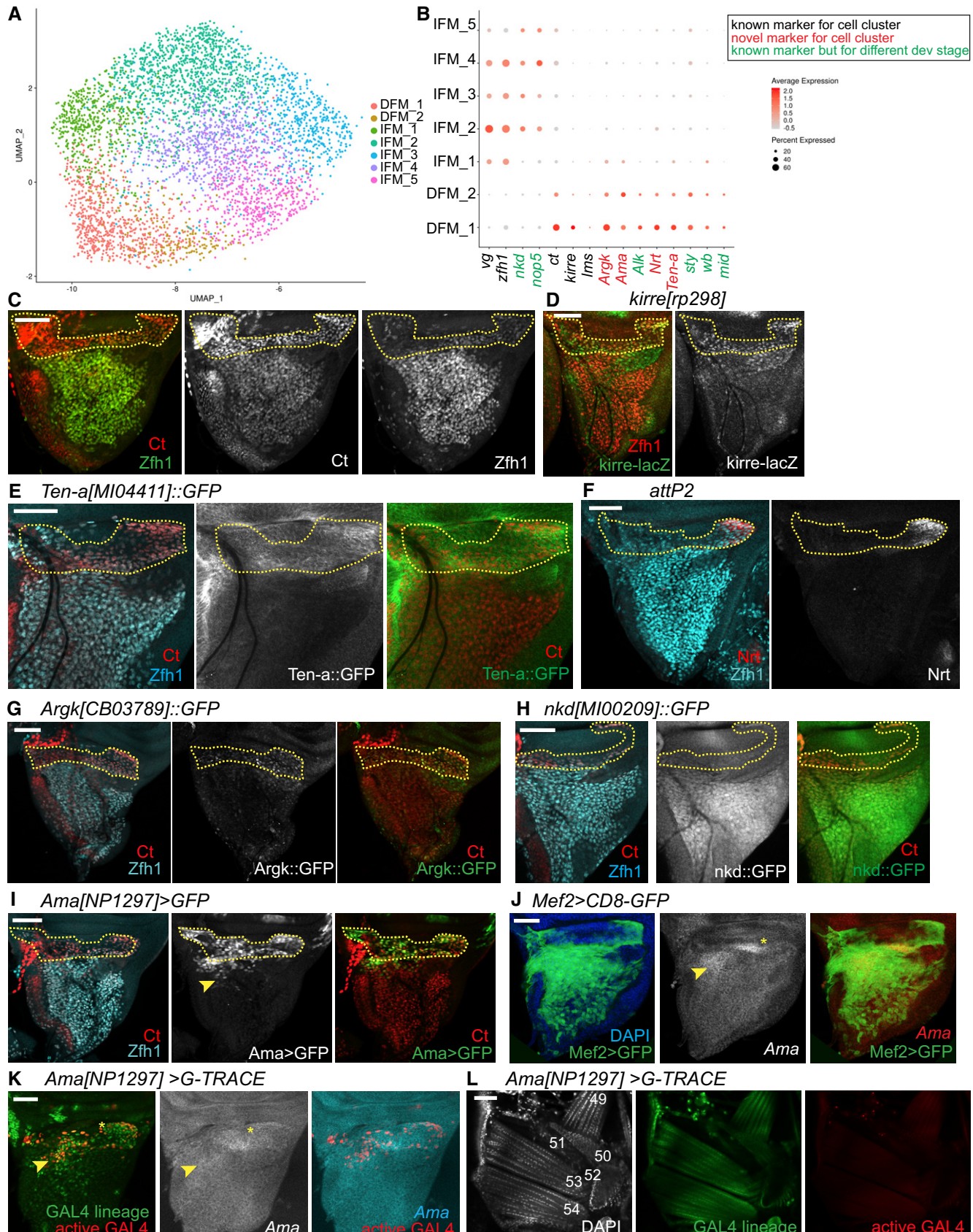

**Figure 3.**

Figure 3. Single-cell transcriptome atlas reveals a large compendium of myoblasts.

A   Subset of 4,544 myoblast cells in UMAP plot of the reference single-cell atlas dataset.
B   Dot plots showing the expression levels of the top variable genes between IFM and DFM myoblasts across 7 clusters of the reference cell atlas dataset.
C–K Confocal single plane images of third instar larval wing discs stained with: (C) anti-Ct (red) and anti-Zfh1 (green), (D) *kirre[rp298]-lacZ* (anti-β-gal, green) and anti-Zfh1 (red), (E) *Ten-a[MI04411]::GFP* (green), anti-Ct (red) and anti-Zfh1 (cyan), (F) anti-Nrt (red) and anti-Zfh1 (cyan), (G) *Argk[CB3789]::GFP* (green), anti-Ct (red) and anti-Zfh1 (cyan), (H) nkd[MI00209]::GFP (green), anti-Ct (red) and anti-Zfh1 (cyan), (I) *Ama[NP1297]>GFP* (green), anti-Ct (red) and anti-Zfh1 (cyan) and (J) *Mef2>CD8-GFP* (green), and *fluorescent in situ hybridization* of *Ama*-RNA probe (white). (K) *Ama[NP1297]>gTRACE* showing the lineage of *Ama-GAL4* (green) and the active *GAL4* (red), and *fluorescent in situ hybridization* of *Ama*-RNA probe (white).
L   Confocal single plane images of adult DFM of *Ama[NP1297]>gTRACE* showing the lineage of *Ama-GAL4* (green) and the active *GAL4* (red) stained with DAPI (white) dorsal right, anterior up. DFM are numbered in white following (Lawrence, 1982).

Data information: Dashed yellow line points to DFM myoblast area. Yellow asterisks and arrowhead indicate *Ama* expression in DFM myoblasts and in a few subset of IFM myoblasts, respectively. Scale bars: 50 μm. Full genotypes: (C) *1151-GAL4, υ-, ω-; +; UAS-mCherry-RNAi*, (D) *kirre[rp298]-nlacZ; +; +*, (E) *y-ω-; Ten-a[MI04411]::GFP*, (F) *y-, υ-; +; P{CaryP}attP2*, (G) *y-, ω-; +; Argk[CBO3789]::GFP*, (H) *y ω; nkd[MI00209]/TM6B, Tb*, (I) *ω-; UAS-GFP; Ama[NP1297]-GAL4*, (J) *ω; UAS-CD8-GFP/Mef2-GAL4* (K-L) *ω-; UAS-gTRACE/+; Ama[NP1297]-GAL4/+*.

Source data are available online for this figure.

Similarly, we used a reporter for *E(spl)m7-HLH* and found that it was expressed in a highly localized manner in the adepithelial layer (Fig 4C) and largely matched the expression pattern of *E(spl)m3-HLH* and *E(spl)mbeta-HLH* (Figs 4D and EV4F). Since these genes are among the top markers of IFM_1-2, we conclude that the cells of these clusters are spatially localized to the anterior region of the presumptive lateral heminotum. Cluster IFM_1 can be distinguished from IFM_2 by the expression of *Sp1*, *Connectin* (*Con*) and *sidestep* (*side*) (Figs 4A and EV4B), and to a lower extent by *Ama*, *Argk* and *wb* (Fig 3B). Both *Con* and *side* are also expressed in embryonic muscles (Nose *et al*, 1992). As we have described above, *Ama* was expressed in a localized manner in a small subset of IFM myoblasts (Fig 3I–K), which largely matches the expression pattern of *E(spl)m3-HLH*, *E(spl)m7-HLH* and *E(spl)m6-BFM*, the markers for IFM_1. Thus, IFM_1 and IFM_2 appear to be spatially localized to the same region of the wing disc notum.

To decipher the relationship between the IFM clusters, we used the computational method Slingshot for inferring cell lineages and pseudotime from the IFM myoblast dataset (Street *et al*, 2018). Cells were ordered from progenitors to differentiated cells (Fig 4E). We identified two lineages: (i) IFM_1 → IFM_2 → IFM_3 → IFM_4 and (ii) IFM_1 → IFM_2 → IFM_3 → IFM_5 (Fig 4F and G), implying that IFM_4 and IFM_5 are quite distinct.

To confirm the *in silico* inference of cell lineages, we performed genetic tracing using gTRACE with the markers for IFM_1 and IFM_2, the *GAL4* drivers *E(spl)m3-HLH* and *E(spl)m6-BFM*. As expected, RFP (active *GAL4*) was found in the specific region for IFM_1-2. By contrast, GFP (*GAL4* lineage) was detected in all myoblasts marked with anti-Ct (Fig 4H and I), thus confirming that the clusters IFM_1-2 contain precursor cells. Our data are in concordance with previous work, in which the AMPs and their myoblast progenies were identified among the pool of myoblasts associated with the wing discs (Gunage *et al*, 2014).

The Notch signalling is thought to maintain cells in an undifferentiated state. The cells in clusters IFM_1 and IFM_2 showed active Notch signalling based on the expression of the *E(spl)* genes, *twi* and *Him*. Accordingly, pseudotime trajectories and tracing experiments identified the clusters IFM_1-2 as AMPs, whereas the cells in the clusters IFM_4-5 correspond to their myoblast progenies and IFM_3 is an intermediate state.

As described above, the cluster IFM_3 likely represented cells transitioning towards differentiation given the low levels of *twi* and

*Him*, and the *in silico* pseudotime inference. Other top markers for this cluster were *Chronologically inappropriate morphogenesis* (*chinmo*), *maternal gene required for meiosis* (*mamo*), *ETS-domain lacking* (*edl*) and *Lamin C* (*LamC*) (Figs 4A and EV4C). *LamC* is also implicated in larval muscle function and leg muscle development (Dialynas *et al*, 2010). We used anti-LamC antibody to examine its expression in the myoblasts by co-staining with anti-Zfh1. Interestingly, although LamC was expressed throughout the adepithelial layer, we noticed that its expression was highly variable as some individual cells had a much higher LamC staining than others (Fig 5A). Two *edl-LacZ* reporters appeared to be expressed in a similar manner, albeit the variation in the expression among the myoblasts were not as pronounced as for LamC (Fig 5B, Appendix Fig S2A). We conclude that IFM_3 comprises cells with high levels of *LamC* and *edl*, and that these cells are distributed throughout the adepithelial layer, unlike the highly localized position for IFM_1 and IFM_2 cells.

Next, we examined the expression of the top markers for the two clusters of differentiating myoblasts, IFM_4 and IFM_5. One of the IFM_4 markers was *hoi-polloi* (*hoip*), which is a regulator of muscle morphogenesis (Johnson *et al*, 2013). Another marker for IFM_4 was *hairy* (*h*), which is also expressed in various embryonic muscles (Fasano *et al*, 1988; Martin *et al*, 2001). As shown in Fig 5C and H was expressed throughout the adepithelial layer, but was largely excluded from the anterior region of the notum where IFM_1 and IFM_2 were located. This is in agreement with the results from our scRNA-seq indicating that *h* marks a unique myoblast cluster (Figs 4A and EV4D).

The cells of IFM_5 cluster expressed components of the extra cellular matrix, the collagen type IV, which is encoded by the genes *Cg25C* and *viking* (*vkg*) (Figs 4A and EV4E). Both *Cg25C* and *vkg* play a role in muscle attachments in the embryo (Borchiellini *et al*, 1996; Junion *et al*, 2007; Hollfelder *et al*, 2014). In the wing disc, the highest expression of *Cg25C* was in the adepithelial layer located at the posterior scutellum, which was determined by staining wing discs of two *Cg25C-lacZ* reporter lines with anti-Zfh1 antibodies (Fig 5D and Appendix Fig S2B). However, the expression of Vkg was detected in myoblasts throughout the adepithelial layer of wing discs, as evidenced by staining two *vkg::GFP* reporter lines (Morin *et al*, 2001) with anti-Ct antibodies (Fig 5E and Appendix Fig S2B).

In summary, scRNA-seq robustly clusters both DFM and IFM myoblasts based on their differentiation status. DFM myoblasts are

grouped into DFM_1 representing undifferentiated myoblasts and DFM_2 containing myoblasts undergoing differentiation. Notably, for IFM myoblasts we discerned two clusters of undifferentiated myoblasts, IFM_1 and IFM_2, a cluster of cells committing to differentiation, IFM_3, and two clusters of differentiating myoblasts, IFM_4 and IFM_5, of which the latter appears to be involved in the formation of basement membrane. Accordingly, our data discriminate between the AMPs (IFM_1-2), an intermediate state (IFM_3),

and their myoblast progeny (IFM_4-5), supporting the previously proposed model of myoblast differentiation (Gunage et al, 2014).

**RNAi screen validates the functional importance of the novel markers identified by scRNA-seq**

Among the list of specific markers for each myoblast cluster (Figs 3B and 4A, Appendix Fig S3, Datasets EV2, EV3, EV5, and

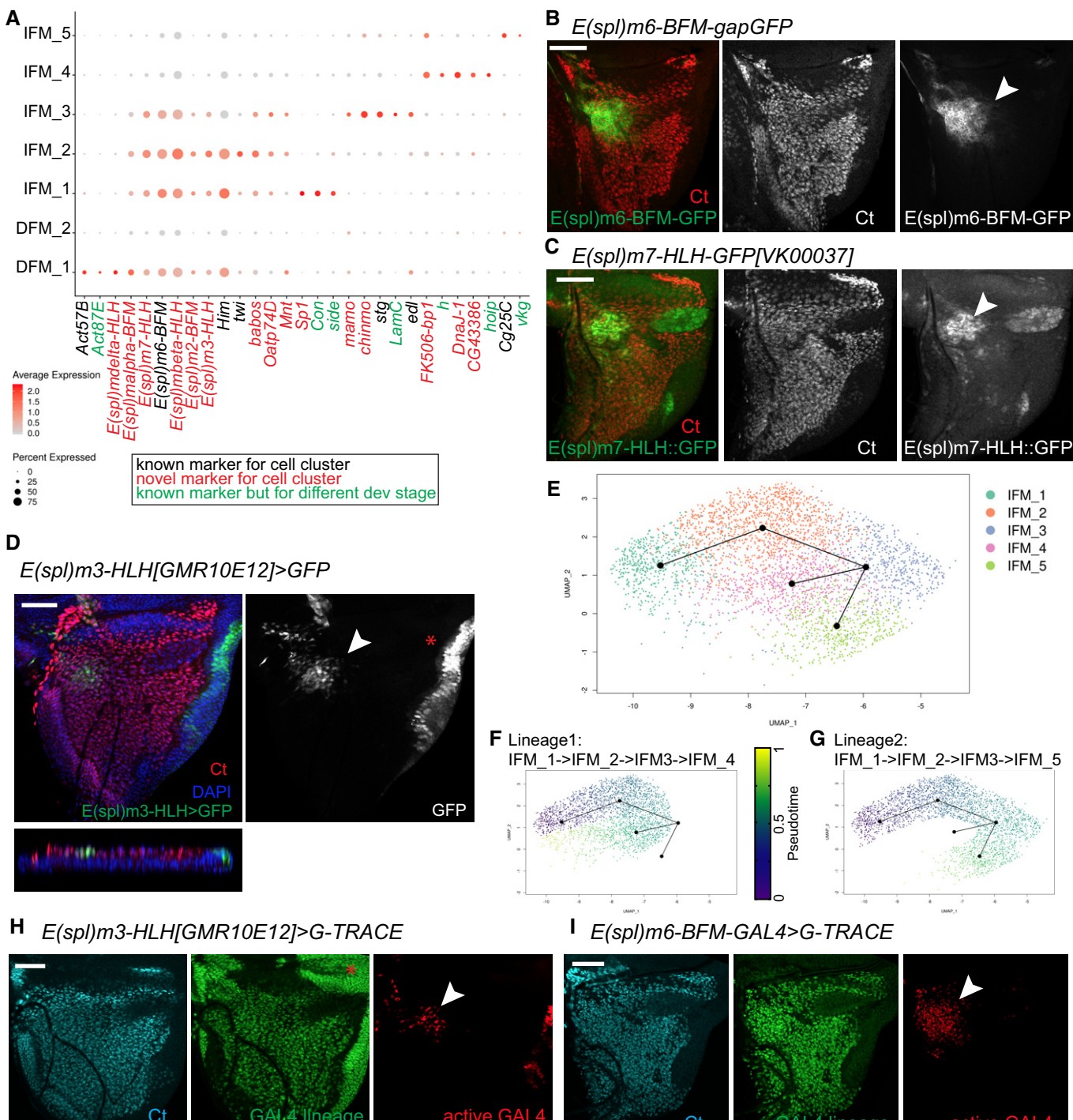

Figure 4.

**Figure 4.  IFM and DFM myoblast clusters uncover different state of differentiation.**

A       Dot plot showing the expression levels of the top variable genes between myoblasts across 7 clusters of the reference cell atlas dataset.

B–D     Confocal single plane images of third instar larval wing discs stained with anti-Ct (red) and (B) *E(spl)m6-BFM-gapGFP* (green), (C) *E(spl)m7-HLH[VK00037]-GFP*
        (green), or (D) *E(spl)m3-HLH[GMR10E12]>GFP* (green).

E–G     Slingshot used for inferring cell lineages and pseudotime among the IFM myoblasts. (E) Two lineages for IFM myoblast clusters visualized in subset of 3,492 cells in
        UMAP plot of the reference single-cell atlas dataset. (F) Pseudotime inference in Lineage 1 is from IFM_1 → IFM_2 → IFM_3 → IFM_4. (G) Pseudotime inference in
        Lineage 2 is from IFM_1 → IFM_2 → IFM_3 → IFM_5.

H, I    Confocal single plane images of third instar larval wing discs stained with anti-Ct (cyan) showing the lineage of *GAL4* (green) and the active *GAL4* (red) in (H) *E(spl)*
        *m3-HLH[GMR10E12]>gTRACE* and (I) *E(spl)m6-BFM-GAL4>gTRACE*.

Data information: White arrowheads point to location of IFM_1-2 cells. Red asterisks indicate expression of *E(spl)m3-HLH* in epithelial cells. Scale bars: 50 µm. Full
genotypes are (B) *w–; UAS-GFP; E(spl)m3-HLH[GMR10E12]-GAL4*, (C) *w–; E(spl)m7-HLH-GFP[VK00037]; +*, (D) E(spl)m6-BFM-gapGFP, (H) *w–; UAS-gTRACE/+; E(spl)m3-HLH*
*[GMR10E12]-GAL4/+*, (I) *w–; UAS-gTRACE, E(spl)m6-BFM-GAL4*.
Source data are available online for this figure.

**A  IFM_3**
    *attP2*

**B  IFM_3**
    *edl[k12907]*

**C  IFM_4**
    *attP2*

**D  IFM_5**
    *Cg25C[A109.1F2]*

**E  IFM_5**
    *vkg[G00454]*

**Figure 5.  Exploring clusters of IFM myoblasts reveals their heterogeneity.**

A–E     Confocal single plane images of third instar larval wing discs stained with (A) anti-LamC (green) and anti-Zfh1 (red), (B) *edl[k12907]-lacZ* (green) and anti-Zfh1
        (red), (C) anti-H (green) and anti-Zfh1 (red), (D) *Cg25C[k00405]-lacZ* (green) and anti-Zfh1 (red), or (E) *ugk[G00454]-GFP* (green) and anti-Ct (red).

Data information: Scale bars: 50 µm. Full genotypes: (A, C) *y–, v–; +; P{CaryP}attP2*, (B) *y–, w; edl[k12907]; +*, (D) *y–, w–; Col4a1[A109.1F2]-lacZ; +*, (E) *w–; vkg-GFP[G00454]; +*

EV6), we noticed a number of novel genes that were not previously linked to the development of the adult flight muscles. To determine the role of these genes in skeletal muscle, we systematically disrupted their function exclusively in the muscle by RNAi (Dietzl *et al*, 2007; Schnorrer *et al*, 2010; Ni *et al*, 2011). First, we scored viability by crossing the publicly available UAS-RNAi transgenes to the pan-muscle *Mef2-GAL4* driver (Fig 6A, Appendix Table S1). If no lethality was observed, adults were tested for their ability to fly as a readout of skeletal muscle function (Fig 6B, Appendix Table S1). If lethality was detected at embryonic or early larval stages, then another driver, *1151*-GAL4, which is specifically expressed in the myoblast cells that give rise to adult skeletal muscles (Anant *et al*, 1998), was used. Both viability and flight ability were assessed (Fig 6C and D, Appendix Table S2). If available, at least two RNAi line stocks targeting the same gene were tested. A total of 47 RNAi lines targeting 34 genes were screened.

Thirteen genes scored in at least one test. One of the top candidates from the screen was *Ama*, which was highly expressed in DFM clusters (Fig 3B and I–K). Its inactivation using either *Mef2-GAL4* or *1151-GAL4* caused early larval and pupal lethality, respectively (Fig 6A and C). The knock down of *Argk*, another top marker for DFM myoblasts, driven by *Mef2-GAL4* led to lethality (Fig 6A; Schnorrer *et al*, 2010). Similarly, the depletion of *nkd* and *nop5* resulted in a flight defect and lethality, respectively (Fig 6A and B). Reducing the expression of *SPARC* and *eEf1alpha2* genes, which are shared markers for IFM and DFM, resulted in early lethality and deficient flight ability, respectively (Fig 6B). The early lethality of *Mef2 > SPARC-RNAi* (Fig 6A; Schnorrer *et al*, 2010) is likely due to its known function in embryonic mesoderm (Martinek *et al*, 2002), as its inactivation with the driver *1151-GAL4*, which is specific for adult skeletal muscles precursors, did not show a muscle-related phenotype (Fig 6C and D). We reasoned that the inactivation of *eEf1alpha2* may exert a broad effect on protein translation and therefore may explain why over 40% of *Mef2 > eEf1alpha2-RNAi* animals were flightless (Fig 6B).

Several new markers for IFM_3, including *chinmo*, *string* (*stg*), *mamo* and *LamC*, seem to be required for muscle formation. The inactivation of *chinmo* using *Mef2-GAL4* and *1151-GAL4* resulted in lethality (Fig 6A and C; Schnorrer *et al*, 2010). As shown in Fig 6E, depletion of *chinmo* by RNAi induced a severe defect on the number of myoblast cells in the adepithelial layer of the wing imaginal discs.

However, the phenotype of *Mef2 > chinmo-RNAi* was not fully penetrant (Fig 6F) and precluded us from further investigating the role of *chinmo*. Another marker for IFM_3 is *stg*, whose inactivation with *Mef2-GAL4* resulted in a 50% reduction in viability and ability to fly (Fig 6A and B). This is likely due to defects in the proliferation of myoblast cells given a well-established role for *stg* in cell cycle. The depletion of *mamo* with either *Mef2-GAL4* or *1151-GAL4* induced lethality at early stages of development (Fig 6A and C). Another hit from the screen was *LamC*, which caused lethality when crossed to *Mef2-GAL4*, but did not show any defect with *1151-GAL4* (Fig 6A, C and D), consistent with its role during larval muscle development (Dialynas *et al*, 2010). Similarly, depletion of *vkg*, a marker for IFM_5, resulted in lethality only with the *Mef2-GAL4* driver (Fig 6A, C and D; Schnorrer *et al*, 2010), in concordance with its known role in muscle attachments in the embryo (Hollfelder *et al*, 2014). Finally, the inactivation of the DFM_1 markers, *E(spl)mdelta-HLH* and *CG11835*, displayed a mild phenotype in flight test (Fig 6B).

## Amalgam is required for the expansion of the myoblast pool in larval wing discs and muscle formation

We selected *Ama* for further analysis as its knockdown displayed a severe muscle-related phenotype (Fig 6A and C), which was validated using two genetic approaches. First, an additional independent *UAS-Ama[GD12733]-RNAi* line phenocopied the viability defect with *Mef2-GAL4* driver (Fig 6G). Second, the lethality of *1151 > Ama[HMS00297]-RNAi* animals (Fig 6C) was partially rescued by overexpressing the *UAS-Ama^{OE}* transgene (Fig 7A). *Ama* encodes an immunoglobulin protein that acts as a cell adhesion molecule during axon guidance in *Drosophila* (Liebl *et al*, 2003). The expression of *Ama* was found preferentially restricted to DFM myoblasts and in very few myoblasts in IFM_1 in the wing disc (Figs 3I–K and EV3C). We began by profiling the proximal *1151 > Ama-RNAi* wing discs by scRNA-seq, as previously done with the wild-type single-cell dataset (Fig 1B). After computational processing, cell clusters were visualized using UMAP (Fig 7B). Cell types, including myoblast, epithelial and tracheal cells, were assigned based on the expression of *zfh1*, *ct*, *Fas3*, *grh* and *trh* (Fig 7C), as in the reference cell atlas (Fig 1E). In total, we captured 3,338 cells across three replicates (Fig EV5A). Strikingly, only 180 of these cells were myoblasts (Fig 7B). In contrast, the wild-type

**Figure 6. Novel markers are functionally important for the development of the flight muscles.**

A–D  The muscle-specific (A, B) *Mef2-GAL4* and (C, D) *1151-GAL4* drivers were crossed to *UAS-RNAi* lines to knockdown the expression of the gene markers for the myoblast clusters. (A, C). Viability test quantified by scoring the percentage of viable animals at each developmental stage, including early pupa (black), pharate (grey) and adult (white), relative to the total number of animals. Data are expressed as stacked bars showing the median with interquartile range ($n \geq 3$ independent biological replicates), unless noted as viable (NQ) and lethal (NQ), which stands for animals that were viable and lethal, respectively, but no quantitative analysis were performed. (B, D). Flight ability scored by quantifying the percentage of flies landing on each section of the column (top, middle and bottom) as indicated on the diagram on the right. Data are expressed as stacked bars. The total number of flies used in each assay was pooled together and displayed next to each bar ($N \geq 6$). n/a stands for non-applicable since animals are not viable.

E  Confocal single plane images of third instar larval wing discs from *Mef2>mCherry-RNAi*, (left panel) and *Mef2>chinmo[HM04048]-RNAi* (right panel) stained with anti–Zfh1 (red) and DAPI (cyan). Scale bars: 50 μm.

F  Quantification of phenotype displayed in (E). Stacked bars showing the percentage of wing discs scored as either normal or as abnormal. Data are expressed as mean ± SEM, $N = 32$ discs per genotype, $n = 3$ independent experiments.

G  Quantification of viability test. Since the *UAS-Ama-RNAi[GD12733]* transgene is inserted on Chromosome X, the genotype of females is *Mef2>Ama-RNAi*, whereas males are *Mef2>+*. Box plot showing the ratio of flies with genotype of interest (females) over control flies (males). The whiskers show 5–95 percentile, and the middle line the median. *$P = 0.014$ (Mann–Whitney test, $N \geq 6$ independent biological replicates). Full genotypes are *w, UAS-Dicer2; +; mCherry-RNAi/Mef2-GAL4* and *w, UAS-Dicer2, UAS-Ama-RNAi[GD12733]; +; Mef2-GAL4*.

Source data are available online for this figure.

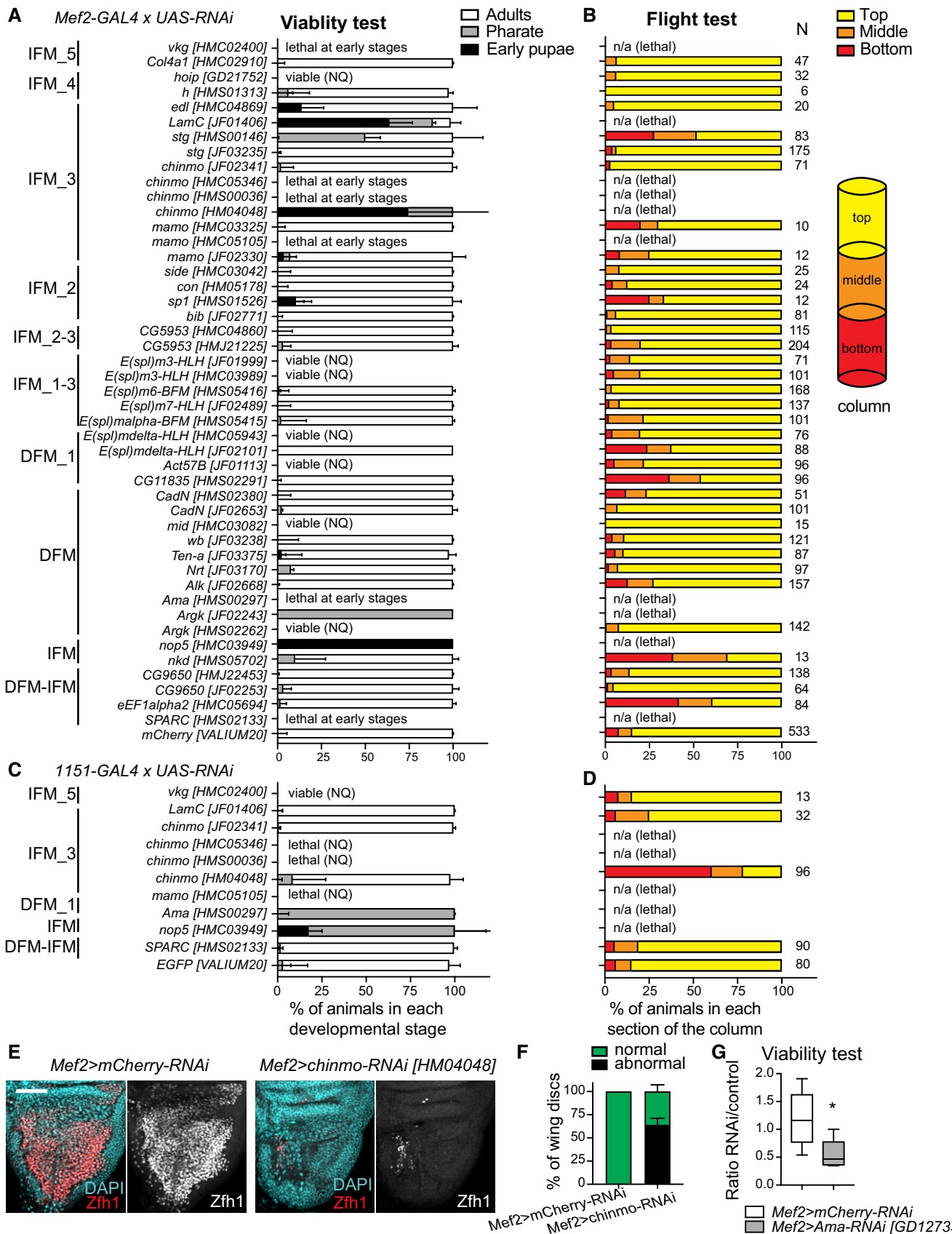

**Figure 6.**

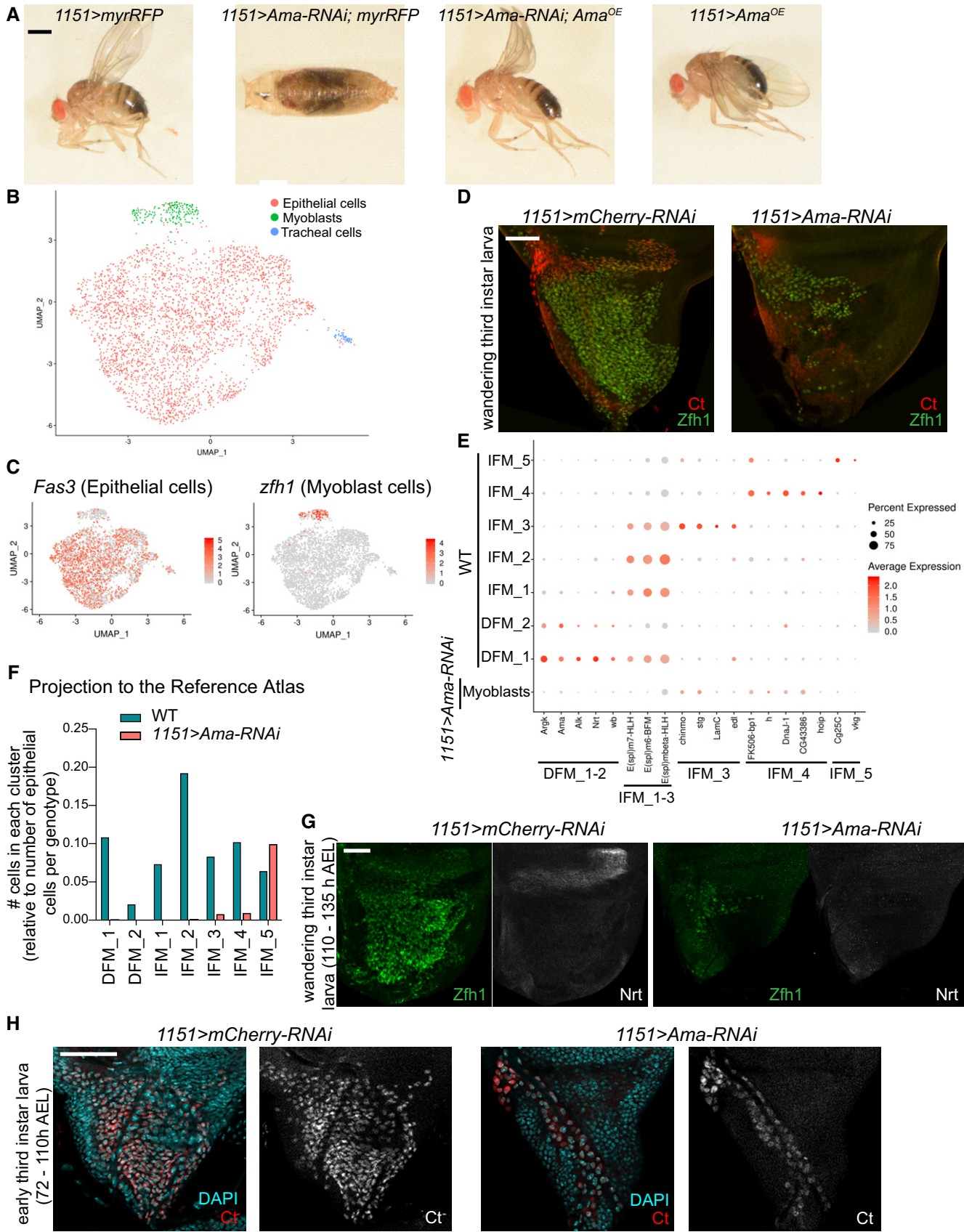

**Figure 7.**

**Figure 7. Amalgam is required for the expansion of myoblast pool in larval wing disc.**

A   Lethality of *1151>Ama-RNAi[HMS00297]* animals at pharate stage was partially rescued by overexpressing the transgene *Ama^OE^*-cDNA. Representative images of adults and pharate pupa are shown. Full genotypes: *1151-GAL4; UAS-myrRFP; +, 1151-GAL4; UAS-myrRFP; UAS-Ama[HMS00297]-RNAi, 1151-GAL4; UAS-Ama^OE^/CyO; UAS-Ama[HMS00297]-RNAi* and *1151-GAL4; UAS-Ama^OE^; +.*

B   Two-dimensional UMAP representation of single-cell RNA-seq *1151>Ama-RNAi* dataset coloured by cell type, including 51 tracheal, 3,107 epithelial and 180 myoblast cells.

C   Average expression level of the genes *Fas3* (left panel) and *zfh1* (right panel) in the *1151>Ama-RNAi* dataset.

D   Confocal single plane images of *1151>mCherry-RNAi* and *1151>Ama-RNAi* wing discs stained with anti-Ct (red) and anti-Zfh1 (green).

E   Dot plots showing the expression levels of the top markers for myoblast clusters identified in the reference cell atlas across the cluster of myoblasts in *1151>Ama-RNAi* and 7 myoblast clusters in control dataset.

F   The myoblast cells in *1151>Ama-RNAi* dataset projected into the reference single-cell atlas by transferring cell-type labels using Seurat. The bar graph represents the total number of cells in each cluster normalized to the total number of epithelial cells per genotype.

G, H   Confocal single plane images of *15111>mCherry-RNAi* and *1151>Ama-RNAi* wing discs stained with anti-Zfh1 (green) and (G) anti-Nrt (white) at wandering third instar larval stage (110–135 h AEL) or (H) DAPI (red) at early third instar larval wing discs (72–110 h AEL).

Data information: Scale bars: 50 μm. Full genotypes: (D, G, H) *1151-GAL4; +; UAS-mCherry-RNAi* and *1151-GAL4; +; UAS-Ama[HMS00297]-RNAi.*

Source data are available online for this figure.

---

atlas contained 4,544 myoblasts among 11,527 cells (Fig 1C), thus indicating that *Ama*-depleted myoblasts are severely underrepresented. To confirm this observation, *1151>Ama-RNAi* late larval wing discs were dissected and stained with anti-Ct and anti-Zfh1 antibodies to visualize myoblasts. In agreement with the scRNA-seq data, there was a drastic reduction in the number of *Ama*-depleted myoblast cells relative to the control (Fig 7D).

To further characterize *Ama*-depleted myoblasts, we examined the myoblast cluster markers from the reference cell atlas (Figs 3B and 4A) for their expression in the myoblast cells of *1151>Ama-RNAi* scRNA-seq dataset (Fig 7E, Dataset EV7). Strikingly, the expression of the DFM markers *Argk, Ama, Alk, Nrt* and *wb*, as well as IFM_1-3 markers, including *E(spl)m6-BFM, E(spl)m7-HLH and E(spl)mbeta-HLH,* was lost in *Ama*-depleted myoblasts. Accordingly, the projection of *1151>Ama-RNAi* cells to the wild-type reference atlas using the cell-type label transfer tool in Seurat revealed that *Ama*-depleted cells were assigned to neither DFM_1-2 nor IFM_1-3 clusters (Fig 7F). By contrast, there was no significant change in the assignment of the epithelial cells of *1151>Ama-RNAi* to the reference atlas (Fig EV5B) or in the expression of the epithelial marker *grh* in these cells (Fig EV5C). Consistent with the loss of expression of the DFM markers in *Ama*-depleted myoblasts, staining of the *1151>Ama-RNAi* wing discs with antibodies against the DFM markers Nrt and Zfh1 revealed a complete lack of DFM myoblasts (Fig 7G).

Although the majority of the *Ama*-depleted myoblast cells were assigned to the IFM_5 cluster (Fig 7F), several IFM_5 markers, including *Cg25C* and *vkg*, were drastically reduced (Fig 7E), implying that the transcriptional programme in *Ama*-depleted myoblasts is dramatically different from the wild-type myoblasts. Interestingly, *Ama*-depleted myoblasts no longer expressed *E(spl)* genes, indicating the loss of the signal to maintain the undifferentiated state. This is additionally reflected in the strongly reduced number of *Ama*-depleted myoblasts remaining on the early third instar larval wing disc, indicating that upon depletion of *Ama*, the myoblast cells fail to undergo a massive expansion that normally occurs at this stage (Fig 7H; Gunage *et al*, 2014). Moreover, since the reduction in the number of the *Ama*-depleted myoblasts (Fig 7D) could be due to either an increased apoptosis or a reduction in cell proliferation, the wing discs were stained with anti-Dcp1, to monitor apoptotic cells, and anti-phosphohistone H3 (pH3), a marker of mitotic cells. We did not observe any increase in these markers in late third instar

discs (Fig EV5D and E). We acknowledge that the low number of myoblasts in *1151>Ama-RNAi* precludes accurate quantification of the phenotypes. We additionally examined the expression of known differentiation markers, including genes involved in myoblast fusion and muscle attachment (Fig EV5F). However, with the exception of *blow* and *mys*, the differentiation markers were not induced as in some wild-type myoblast clusters. This indicates that *Ama*-depleted myoblasts are unable to proliferate and blocked from undergoing proper differentiation.

To determine whether the remaining *Ama*-depleted myoblasts are competent to form muscles, we examined the formation of DFM and IFM in *1151>Ama-RNAi* at early stages of pupal development. Although myoblasts properly migrated and were detected at early pupae stages in *1151>Ama-RNAi*, no developing DFM were visible at 40 h APF (Fig 8A). Likewise, no developing IFM were detected in *1151>Ama-RNAi* at 16 h APF (Fig 8B) nor at 20 h APF (Appendix Fig S4). Notably, larval oblique muscles, which are used as templates for IFM formation, were not found, suggesting that these may not have escaped histolysis. To exclude the possibility that *Ama*-depleted myoblasts are simply delayed in development and form muscle later, we examined the adult skeletal muscles in *1151>Ama-RNAi* pharate animals at 96 h APF. Thoracic sections were stained with Phalloidin and antibodies against Kettin and PS-integrin to visualize the myofibril structure of DFM and IFM by immunofluorescence. Strikingly, DFM were completely missing in *1151>Ama-RNAi* in comparison to control animals (Fig 8C). Likewise, IFM muscles were detected in neither transverse nor sagittal thoracic sections (Fig 8D, top and bottom panels, respectively).

From these data, we conclude that *Ama*-depleted myoblasts are severely impaired in their proliferation capacity and that the transcriptional programme of the remaining *Ama*-depleted myoblasts is significantly perturbed. Further, it appears the remaining *Ama*-depleted myoblasts are unable to differentiate, resulting in a block in myoblast fusion in the developing pupa and complete loss of both the IFM and DFM.

## Discussion

The myoblast cells associated with the larval wing discs give rise to two distinct types of adult flight muscles, the fibrillar IFM and the tubular DFM, with distinct physiology, size, contractile properties

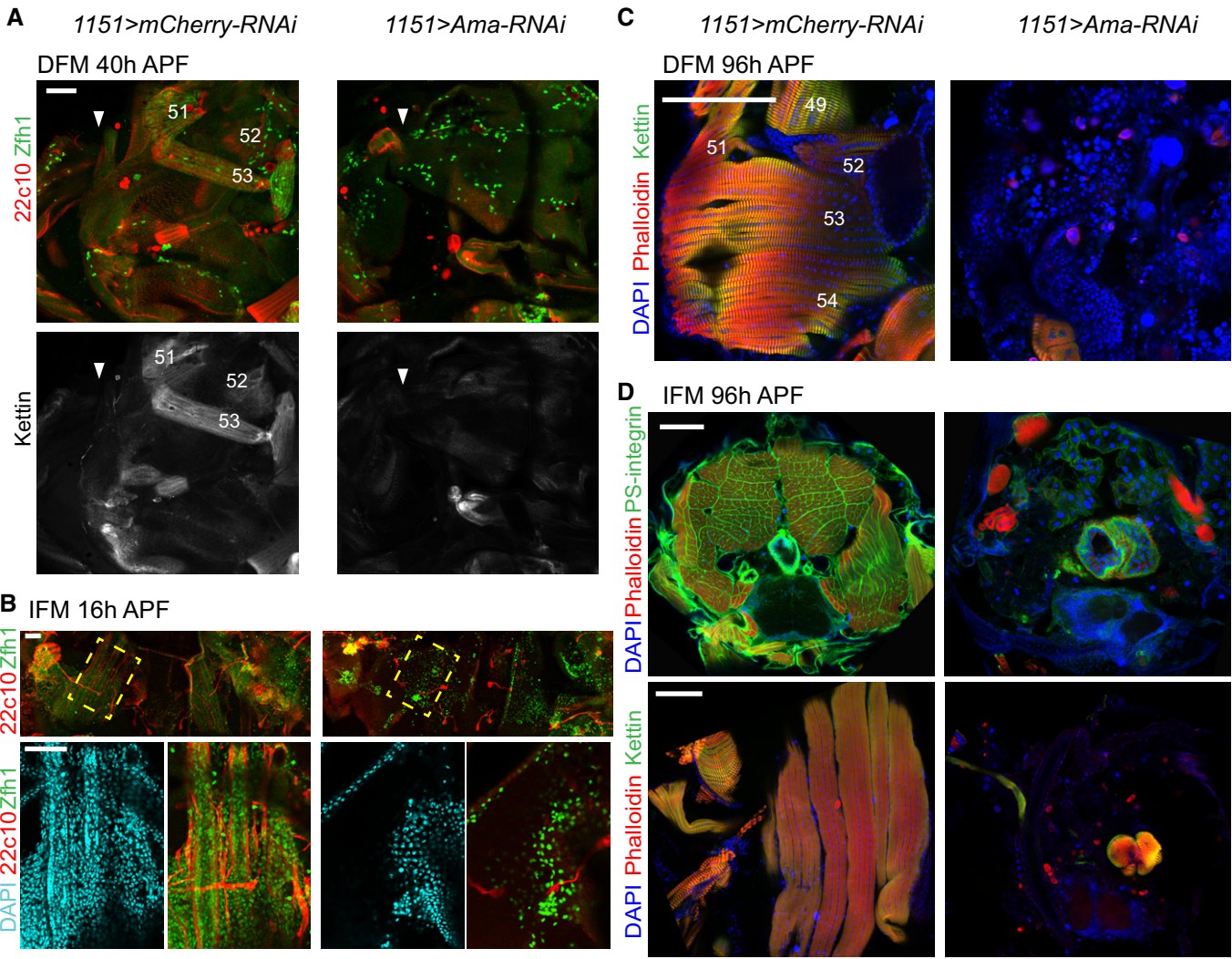

**Figure 8. The loss of Amalgam impairs the formation of both IFM and DFM at early stages of developing pupa.**

Confocal single plane images of *1151>Ama-RNAi* animals and *1151>mCherry-RNAi*.

A   Forming DFM at 40 h APF stained with anti-Zfh1 (green), anti-Futsch (22c10, red) and anti-Kettin (white). White arrows point to the wing hinge, wings pointing left, anterior up.

B   Forming IFM (DLM) at 16 h APF, stained with anti-Futsch (22c10, red), anti-Zfh1 (green) and DAPI (cyan). Yellow-dashed box indicates magnified area (bottom panel). Anterior up.

C   DFM at 96 h APF stained with Phalloidin (red), anti-Kettin (green) and DAPI (blue). Anterior up, dorsal right. DFM are numbered in white as in (Lawrence, 1982).

D   IFM at 96 h APF, transverse section (top) stained with anti-PS-integrin (green), Phalloidin (red) and DAPI (blue), dorsal up; sagittal section (bottom) stained with anti-Kettin (green), Phalloidin (red) and DAPI (blue), anterior up, dorsal right.

Data information: Scale bars: 50 μm (A, B), and 100 μm (C, D). Full genotypes: *1151-GAL4; +; UAS-mCherry-RNAi* and *1151-GAL4; +; UAS-Ama[HMS00297]-RNAi*.

and metabolic characteristics. To account for these differences, the myoblasts undergo complex diversification during development that culminates in the expression of fibre-type-specific structural genes, among others. However, it is not clear when the major differences in their intrinsic transcriptional programmes arise. With the advancement in single-cell technologies, it is now feasible to dissect the transcriptomes of individual cells and accurately identify cell types, cell states, gene signatures and major genetic drivers of developmental programmes. In this study, we performed single-cell RNA-seq to build a reference atlas of 4,544 myoblast cells associated with the third instar larval wing disc notum. The atlas represents

approximately 1.8× cellular coverage based on the estimated number of 2,500 myoblasts per wing disc at this developmental stage (Gunage *et al*, 2014). By querying the cell atlas, we dissected cell heterogeneity across myoblast clusters and explored early events in establishing muscle diversity. Here, we report three main findings.

First, we show that in the third instar larva, the IFM and DFM myoblasts have distinct transcriptional programmes. Our data are in agreement with previous studies showing that the divergence of the myoblast cells correlates with the differential expression of *vg* and *ct* (Sudarsan *et al*, 2001). It has been suggested that Wg emanating

from the wing disc epithelium is required for the maintenance of Vg expression and thus helps to spatially subdivide myoblasts into DFM and IFM myoblasts (Sudarsan *et al*, 2001). One implication of our results is that the divergence is not merely because DFM and IFM myoblasts are in distinct regions of the notum defined by the pattern of Wg expression, but rather due to the extensive differences in the transcriptional programmes between IFM and DFM myoblasts that are not limited to *vg* and *ct*. Interestingly, such transcriptional changes occur prior to expression of fibre-type-specific genes that are thought to distinguish fibre types at the molecular level (Schiaffino & Reggiani, 2011). At least two distinct regulatory mechanisms mediated by Salm, Exd and Hth govern fibre-type fate by regulating the expression of components of myofibrillar structure in later myogenesis (Schönbauer *et al*, 2011; Bryantsev *et al*, 2012). The differences in gene expression in the proliferating myoblasts that we report here raise the question of what other factors are contributing to the establishment of muscle diversity at early stages of myogenesis.

Our work identifies a number of cell-type-specific markers that distinguish DFM myoblasts from IFM myoblasts and were largely missed in previous studies using bulk RNA-seq (Spletter *et al*, 2018; Zappia *et al*, 2019). We acknowledge that this list is likely incomplete due to a low depth of sequencing in Drop-seq. Interestingly, the list of markers includes *trol*, *nkd*, *Alk*, *sty*, *wb*, *mid*, *Con*, *side*, *LamC*, *h*, *hoip* and *vkg*, which are also involved in the formation of embryonic muscles, thus indicating that these genes are reused during the development of the adult skeletal muscle.

Since the myoblasts are associated with the epithelial cells of the wing disc, we also recovered 6,711 epithelial cells and 272 tracheal cells. Although this was not the goal of this work, we mapped 17 epithelial cell clusters to the wing disc fate map and two tracheal cell clusters. Together with two recent studies (Bageritz *et al*, 2019; Deng *et al*, 2019), our work provides a valuable resource for the *Drosophila* community as a part of The Fly Cell Atlas initiative (https://flycellatlas.org/).

The second conclusion of our work is that the populations of IFM and DFM myoblasts are highly heterogeneous, as we identified five IFM and two DFM clusters that represent cells at various states of differentiation. Consistent with the well-documented role of the Notch pathway in the formation of IFM, we found clusters of myoblasts expressing *E(spl)* genes, which are indicative of Notch activity, and clusters with low expression of *E(spl)* genes, which likely represent more differentiated myoblasts. Such interpretation is supported by pseudotime trajectories and genetic tracing experiments that allowed us to follow the lineage of cells expressing *E(spl)* genes *in vivo*. Thus, we infer that the clusters IFM_1-2 mostly contain AMPs, whereas the myoblasts in the clusters IFM_4-5 are likely their progenies. This is consistent with clonal analyses done in a previous report (Gunage *et al*, 2014). Hence, scRNA-seq reveals the temporal progression of muscle precursor cells across distinct states of differentiation.

Interestingly, the expression of some downstream effectors of the Notch pathway, such as *E(slp)-mdelta*, differs between the IFM and DFM myoblasts, suggesting that although Notch is the main driving force in the regulatory network, the output is likely modulated by other myoblast-type-specific genetic cues. Noteworthy, the expression of *E(spl)m3-HLH*, *E(spl)m7-HLH*, *E(spl)mbeta-HLH* and *E(spl)m6-BFM* (Figs 4B–D and EV4F; Lai *et al*, 2000), which are the top

markers for the clusters IFM_1 and IFM_2, is localized within the adepithelial layer near the anterior region of the presumptive lateral heminotum. These data suggest that, at least, the populations IFM_1 and IFM_2 are spatially restricted. This is in contrast to other clusters, such as IFM_3, which are largely distributed throughout the adepithelial layer. Curiously, in addition to *E(spl)* genes, the cluster IFM_1 expresses several DFM markers, such as *Ama*, *Argk*, and *wb*, and we confirmed the expression of *Ama* in a subset of myoblasts in IFM_1 cluster using three approaches: FISH, *Ama-GAL4* reporter and genetic tracing experiments. Finally, we disfavour the explanation that the cluster IFM_1 contains cell doublets of IFM and DFM myoblasts as this cluster shows *vg* and low *ct* expression, an IFM hallmark (Sudarsan *et al*, 2001).

Third, our work provides a framework for leveraging scRNA-seq to identify novel genes that are functionally important in a particular biological process. Concordantly, several novel muscle genes identified here as markers for myoblasts also scored in a large-scale screen for genes involved in muscle morphogenesis and function (Schnorrer *et al*, 2010). Using the list of marker genes as an entry point for a candidate RNAi screen, we discovered *Ama*, whose inactivation caused severe muscle defects. Ama was shown to act as a ligand for the cell adhesion molecule Nrt during axon guidance in *Drosophila* embryogenesis (Fremion *et al*, 2000). It has been proposed that Ama facilitates Nrt-mediated adhesion by functioning as a linker between two Nrt-expressing cells. Intriguingly, both Nrt and Ama are highly expressed in the DFM myoblasts, raising the possibility that Ama may similarly interact with Nrt in the myoblasts to regulate adhesion. We found that depletion of Ama led to a severe reduction in the number of myoblasts, which is most likely due to the failure to undergo expansion during larval stages. Interestingly, Ama was shown to regulate the expression of Cyclin E through the Hippo pathway (Becker *et al*, 2016), which may explain the proliferative defects we report here. Strikingly, the remaining *Ama*-depleted myoblasts are unable to form myotubes, which is likely caused by the highly abnormal transcriptional profile of the *Ama*-depleted myoblasts, as revealed by scRNA-seq. Thus, the failure to properly execute the myogenic transcriptional programme may underlie muscle defects upon the loss of Ama function.

One limitation of this approach is that it may fail to identify low expressed genes, or conversely, the *UAS-RNAi* lines may not be robust enough to display a phenotype with the specific muscle driver. Therefore, the list of candidates that we report here is likely to be incomplete. Nevertheless, this strategy can be applied to other scRNA-seq datasets to address the functional significance of novel marker genes identified as a part of the scRNA-seq computational pipeline. Collectively, our work illustrates the power of combining single-cell genomics with genetic approaches and cell lineage tracing experiments to address important questions in developmental biology.

## Materials and Methods

### Fly maintenance and stocks

All lines used here are listed in Appendix Table S3 (Morin *et al*, 2001; Dietzl *et al*, 2007; Evans *et al*, 2009; Ni *et al*, 2011; Venken *et al*, 2011; Jenett *et al*, 2012; Aradhya *et al*, 2015; Nagarkar-Jaiswal

*et al*, 2015; Kudron *et al*, 2018). All fly crosses were kept at 25°C in vials containing standard cornmeal-agar medium.

### Ama^OE transgene

The clone *UAS-Ama* C-terminal tagged with FLAG-HA (UFO01101, Drosophila Genomics Resource Center DGRC 1621050), which contains the gene selectable marker *white* and the site *attB*, is from the Universal Proteomics Resources, a Berkeley Drosophila Genome Project (BDGP) (https://www.fruitfly.org/EST/proteomics.shtml).

The *y[1] w[1118]; PBac{y[ + ]-attP-3B}VK00002* line (Bloomington Drosophila Stock Center, BDSC 9723) with an estimated Cyto-Site 28E7 (Chromosome *2*) was used for PhiC31 integrase-mediated transgenesis of the *UAS-Ama^OE* construct. *Drosophila* embryo injection services from BestGene were used (http://www.thebestgene.com/).

### Fly viability assay

Adult flies able to eclose from the pupal case, dead pharate and dead pupae were counted to score the percentage of flies that made it to each developmental stage. Pupal developmental stages were determined by observing biological markers of metamorphosis. At least 50 flies per genotype were screened in total (unless lethal at early stages), with a minimum of two independent experiments.

Viability assays of *Mef2>Ama-RNAi [GD12733]* along with matching control were done in the background of *UAS-Dicer2* to enhance the RNAi machinery. Development was set at 29°C to increase *GAL4* activity. Since RNAi is inserted in Chromosome X, and males carrying RNAi were used to set up the crosses, we scored the number of female (genotype of interest) and number of male (genotype control) from segregating populations. The ratio female/male (genotype of interest/control) was calculated and plotted.

### Flight test

Males no older than a week were collected on $CO_2$ and kept at 25°C for at least 24 h for recovery, and then at room temperature for another hour for acclimation. Flies were then flipped into a 2-l graduated cylinder lined with a 432 mm high piece of paper coated in mineral oil. Flies landed on the paper at different heights depending on their ability to fly. A picture of the unfurled paper was then taken, and the landing spot for each fly was unbiasedly scored as one of the three sections (top, middle or bottom) by an ImageJ plugin (script available upon request). Frequencies were then analysed and plotted. Experiments were carried out at least twice for genotypes presenting a phenotype, at least 25 flies per genotype (except for *Mef2 > con[HM05178]-RNAi, Mef2 > h[HMS01313]-RNAi, Mef2 > mamo[HMC03325]-RNAi, Mef2 > mamo[JF02330]-RNAi, Mef2 > sp1[HMS01526]-RNAi, Mef2 > mid[HMC03082]-RNAi, Mef2 > nkd [HMS05702]-RNAi, 1151 > vkg[HMC02400]-RNAi*).

### Dissection and immunofluorescence

#### Imaginal wing discs
Larvae were dissected in phosphate-buffered saline (PBS) pH 7.4 and immediately fixed with 4% formaldehyde in PBS for 30 min (15 min whenever anti-Ct antibody was used), permeabilized twice

in 0.3% Triton X-100-PBS for 10 min and blocked in 10% normal donkey serum (NDS) 0.1% Triton X-100-PBS for 1 h. Samples were incubated in primary antibodies overnight at 4°C in 10% NDS 0.1% Triton X-100-PBS, washed for 5 min three times with 0.1% Triton X-100-PBS, then incubated for 1 h with secondary antibodies and dyes in 10% NDS 0.1% Triton X-100-PBS. Samples were washed with 0.1% Triton X-100-PBS five times for 5 min.

#### Developing pupal flight muscles
Thoraces were dissected as in (Weitkunat & Schnorrer, 2014), fixed for 15 min with 4% formaldehyde in PBS, permeabilized in 0.3% Triton X-100-PBS three times for 20 min and blocked in 10% NDS 0.1% Triton X-100-PBS for 120 min. Then primary antibody was incubated overnight at 4°C in 10% NDS 0.1% Triton X-100-PBS, four 25 min washes in 0.1% Triton X-100-PBS, secondary antibody was incubated for 120 min in 10% NDS 0.1% Triton X-100-PBS, and four 15 min washes in 0.1% Triton X-100-PBS.

#### Adult and pharate flight muscles
For transverse sections, flies were snap-frozen in liquid nitrogen, cut twice with a razor and fixed for 2 h in 4% formaldehyde in relaxing buffer (20 mM phosphate buffer, pH 7.0; 5 mM $MgCl_2$; 5 mM EGTA). For the sagittal IFM and DFM sections, thoraces were cut from the animals, incubated in relaxing buffer for 15 min, fixed for 30 min in 4% formaldehyde in relaxing buffer, cut through the appropriate sagittal plane with a Sharpoint 22.5° Stab Knife (ID#72-2201) and then fixed for an additional 15 min as described in (Schnorrer *et al*, 2010). All subsequent solutions contained 0.3% Triton X-100 for transverse and sagittal IFM and 0.5% Triton X-100 for DFM. Four 15 min washes for transverse and three 10 min for sagittal sections, 2-h incubations in blocking solution (PBS + 2% bovine serum albumin (BSA) + Triton 100-X). Sections were incubated with primary antibodies in blocking solution overnight at 4°C, washed four times with PBS solutions, 15 min for transverse and 10 min for sagittal sections. Secondary antibody incubations were 2 h long in 0.3% Triton X-100 10% NDS-PBS for transverse and sagittal IFM, and 0.5% Triton X-100 2% BSA in PBS for DFM sections. Finally, all sections were washed four times for 10 min.

Primary antibodies, secondary antibodies and dyes used in this work are listed in Appendix Table S4. After washing off the secondary antibodies, all samples were stored in 0.5% propyl gallate 50% glycerol at 4°C until mounted on glass slides.

### Fluorescent *in situ* hybridization

Custom probes were designed against Ama-RA coding sequence by utilizing the Stellaris. FISH Probe Designer v4.2 (Biosearch Technologies, Inc., Petaluma, CA) available online at www.biosearchtech.com/stellarisdesigner. The following parameters were used Oligo length = 18; min. spacing length = 2; masking level = 5. The designer generated 38 probes.

The wing discs were hybridized with the Ama Stellaris FISH Probe set labelled with Quasar 670 fluorophores dye (Biosearch Technologies, Inc.), following the manufacturer's instructions available online as "Protocol for D. Melanogaster Wing Imaginal Discs" at https://biosearchassets.blob.core.windows.net/assets/bti_custom_stellaris_drosophila_protocol.pdf.

## Microscopy

All images were taken with a Zeiss LSM Observer.Z1 laser-scanning confocal microscope, using 10×/0.30, 20×/0.8, 40×/1.20 and 100×/1.45 objectives. Pinhole was kept at 1 AU, laser and gain was kept consistent within experiments (e.g. keeping the same settings for control and knockdown). For z-stacks, the optimal slice size suggested by the microscope software was used. Orthogonal views were taken from the z-stacks using ImageJ. Only one representative image per experiment is shown.

## Software

The software used were NIH ImageJ 1.52k5 https://imagej.nih.gov/ij/for image visualization, Adobe Photoshop CS6 version 13.0.6 https://www.adobe.com/products/photoshop.html for image editing, GraphPad Prism version 8.0.1 https://www.graphpad.com create all graphs, Adobe Illustrator CC 23.0.2 https://www.adobe.com/products/illustrator.html for figure editing.

## Single-cell preparation

Wandering third instar larvae *1151-GAL4* samples and *1151>mCherry-RNAi* samples were harvested between 110 and 135 h AEL at 25°C. Tissue was dissociated into single-cell suspension as in Ariss *et al* (2018). Briefly, wing discs were dissected in cold PBS1×, pouch was manually removed with microblade. Notum and hinge were collected and processed for dissociation in a final concentration of 2.5 mg/ml Collagenase (Sigma #C9891) and 1× trypsin (Sigma #59418C) in Rinaldini solution. The microcentrifuge tube was horizontally positioned on a shaker set at 225 rpm for 20 min at RT. Cells were washed twice and resuspended in 0.04% BSA-PBS. Cell viability and concentration were assessed by staining cells with Trypan blue and counting using a hemocytometer.

## Sample preparation for scRNA-seq

For Drop-seq, we followed the protocol version 3.1 (12/28/15) as in Macosko *et al* (2015) posted in http://mccarrolllab.org/dropseq/ with the following modifications. The lysis buffer contained 0.4% Sarkosyl (Sigma). The number of cycles in the PCR step post-exonuclease is 4 and then 12. The cDNA post-PCR was purified twice with 0.6× Agencourt AMPure XP (Beckman Coultier). The tagmented DNA for sequencing was purified twice: first using 0.6× and the second time using 1× Agencourt AMPure XP (Beckman Coultier).

## High-throughput sequencing

Quality control of both amplified cDNA and sequencing-ready library was determined using Agilent TapeStation 4200 instrument. All Drop-seq libraries were sequenced on a NextSeq instrument (Illumina). Sequencing was done at the University of Illinois at Chicago Sequencing Core (UICSQC).

## Preprocessing raw datasets of scRNA-seq

The Drop-seq samples were processed for read alignment and gene expression quantification following Drop-seq cook-book (version 1.2 Jan 2016)7 (http://mccarrolllab.com/dropseq/; Macosko *et al*, 2015). We used STAR aligner to align the reads against *Drosophila melanogaster* genome version BDGP6 (Ensembl version 90). Quality of reads and mapping were checked using the program FastQC (https://www.bioinformatics.babraham.ac.uk/projects/fastqc/). Digital Gene Expression (DGE) matrix data obtained from an aligned library were done using the Drop-seq program DigitalExpression (integrated in Drop-seq_tools-1.13). Number of cells that were extracted from aligned BAM file is based on knee plot which extracts the number of reads per cell, then plot the cumulative distribution of reads and select the knee of the distribution.

## scRNA-seq data analysis

The packages Seurat (version 3.0.0) and R (version 3.5.3) were used to analyse datasets and to generate plots. We followed the standard tutorial instructions from the Seurat website (https://satijalab.org/seurat/; Stuart *et al*, 2019). First, the gene expression matrices were subjected to an initial quality control analysis. Low-quality cells were filtered out using 200 and 2,500 gene/cell as a low and top cut-off, respectively, and min.cell = 5. Additionally, markers of cellular stress, such as mitochondrial content, heat shock protein and 28sRNA, were set to less than 10, 6 and 1.5% reads per cell, respectively, to filter out low-quality cells. A global-scaling normalization method, which normalized the gene expression values for each cell by the total expression, was used. Then, data were multiplied by a scale factor (10,000) and log-transformed. Next, linear transformation (or scaling) was applied. The expression of each gene was shifted, so that the mean expression across cells was 0, and was scaled, so that the variance across cells was 1. The four variables mentioned earlier were regressed out when scaling.

Next, an integrative analysis was done to generate the reference cell atlas for the wild-type wing discs by determining the anchor points in each dataset using Seurat pipeline. The top 2,000 variables genes were used as input for PCA analysis to run linear dimensional reduction. Cells were clustered using a graph-based method. The first 30 principle components were selected to run non-linear dimensional reduction (UMAP) using granularity 2.5 for reference atlas and 1.5 for *1151 > Ama-RNAi* dataset. Clusters were visualized with UMAP (Appendix Fig S1A). Clusters of cells that were not evenly distributed among replicates, i.e. more than 50% bias for one or two samples, which indicates a batch effect between samples, were removed from the analysis to minimize technical or dissection bias (red arrows, Appendix Fig S1B). Also, clusters expressing markers of cellular stress and dying cells, such as heat shock protein, high content of mitochondrial genes and rRNA, were removed from the analysis. The number of genes and transcripts per cluster were monitored to find potential multiplets. Finally, clusters were removed if specific markers for both epithelial and myoblast cells were co-expressed. Then, dataset was reanalysed. The analysis was performed only on the cell barcodes as listed in Dataset EV8. The dimensions of the final datasets are 11,874 genes by 11,527 cells for the reference atlas and 8,483 genes by 3,338 cells for Ama-knockdown.

The cluster-specific markers were determined by calculating differential expression using non-parametric Wilcoxon rank sum test. In the reference atlas, two IFM clusters originally showed same top

markers. Thus, to prevent an artefact of overclustering, these two clusters were labelled as IFM_2.

Flybase (http://flybase.org) was used to browse and explore the functional genetic data for the biomarkers in each cell cluster (Thurmond *et al*, 2019).

Cell lineage and pseudotime were inferred using Slingshot (Street *et al*, 2018). Only the IFM clusters of the wild-type reference dataset were selected for the analysis.

## Data availability

Drop-seq scRNA-seq data have been deposited in the NCBI Gene Expression Omnibus database (GEO, https://www.ncbi.nlm.nih.gov/geo/) and are accessible through the accession number GSE138626 (https://www.ncbi.nlm.nih.gov/geo/query/acc.cgi?acc=GSE138626). Raw and processed data were also deposited in the European Bioinformatics Institute ArrayExpress (https://www.ebi.ac.uk/arrayexpress/).

**Expanded View** for this article is available online.

## Acknowledgements

We thank T.V. Orenic for helping with the assignment of the epithelial clusters, R.M. Cripps, M. Spletter, F. Schnorrer, I. K. Hariharan for helpful discussions and Anna Barque for helping with crosses. We are grateful to R.M. Cripps, K. Jagla, the Bloomington Drosophila Stock Center (supported by NIH grant P40OD018537), the Vienna Drosophila Resource Center, the TRiP at Harvard Medical School and the KYOTO Stock Center (DGRC) at Kyoto Institute of Technology for fly stocks; the Babraham Institute and the Developmental Studies Hybridoma Bank (DSHB) for antibodies; the Drosophila Genomics Resource Center (DGRC, supported by NIH grant 2P40OD010949) and the Berkeley Drosophila Genome Project (BDGP) for *Ama*$^{OE}$-cDNA construct; and to Flybase for online resources on the Database of Drosophila Genes & Genomes. We thank the University of Illinois at Chicago Sequencing Core (UICSQC) for sequencing Drop-seq samples. This work was supported by NIH grant R35GM131707 (to M.V.F.).

## Author contributions

MPZ and MVF conceived the project, designed the experiments, analysed data, and wrote the manuscript. MPZ performed Drop-seq on the wing discs of wild type and *Ama*-RNAi and analysed all the single-cell RNA-seq datasets. LdC and HJ generated all immunofluorescences and the data from the screen. LdC also helped MPZ with figure preparation. MMA performed Drop-seq on the wild-type wing discs, *Ama* mRNA FISH in wing discs, and made transgenic *UAS-Ama*$^{OE}$. ABMMKI helped with the bioinformatics data analysis of high-throughput sequencing datasets. All authors reviewed and edited the manuscript.

## Conflict of interest

The authors declare that they have no conflict of interest.

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
