## [Review Process File · EMBO Reports]

A cell atlas of adult muscle precursors uncovers early events in fiber-type divergence in *Drosophila*

Maria Zappia, Lucia de Castro, Majd Ariss, Holly Jefferson, Abul Islam, and Maxim Frolov
DOI: [10.15252/embr.201949555](https://doi.org/10.15252/embr.201949555)

Corresponding author(s): Maxim Frolov (mfrolov@uic.edu)

Review Timeline:

Submission Date:	29th Oct 19
Editorial Decision:	12th Dec 19
Revision Received:	18th May 20
Editorial Decision:	10th Jul 20
Revision Received:	12th Jul 20
Accepted:	24th Jul 20

Transaction Report:

Dear Dr. Frolov

Thank you for the submission of your research manuscript to our journal and for your patience while it was under review. We have now received the full set of referee reports that is copied below.

Your manuscript was reviewed by two *Drosophila* muscle experts (referee 1 and 2) and by one reviewer with expertise in single-cell 'omics' and developmental gene expression (referee 3). As you will see, all three referees acknowledge that the findings are potentially interesting. However, all three referees also point out several technical concerns and have a number of suggestions for how the study should be strengthened. It will be essential to validate new marker genes by staining for gene expression either using antibody staining or in situ hybridization and to improve data presentation and description. Importantly, the functional data on the role of *Ama* need to be substantiated with further experiments including the use of additional RNAi lines.

Given these constructive comments, we would like to invite you to revise your manuscript with the understanding that the referee concerns (as detailed above and in their reports) must be fully addressed and their suggestions taken on board. Please address all referee concerns in a complete point-by-point response. Acceptance of the manuscript will depend on a positive outcome of a second round of review. It is EMBO reports policy to allow a single round of revision only and acceptance or rejection of the manuscript will therefore depend on the completeness of your responses included in the next, final version of the manuscript.

Revised manuscripts should be submitted within three months of a request for revision; they will otherwise be treated as new submissions. Please contact us if a 3-months time frame is not sufficient for the revisions so that we can discuss the revisions further.

- 1) A data availability section providing access to data deposited in public databases is missing (if relevant).
- 2) Your manuscript contains error bars based on $n=2$. Please use scatter blots showing the individual datapoints in these cases. The use of statistical tests needs to be justified.

2) individual production quality figure files as .eps, .tif, .jpg (one file per figure).

Please download our Figure Preparation Guidelines (figure preparation pdf) from our Author Guidelines pages

<https://www.embopress.org/page/journal/14693178/authorguide> for more info on how to prepare your figures.

3) a .docx formatted letter INCLUDING the reviewers' reports and your detailed point-by-point

responses to their comments. As part of the EMBO Press transparent editorial process, the point-by-point response is part of the Review Process File (RPF), which will be published alongside your paper.

4) a complete author checklist, which you can download from our author guidelines (). Please insert information in the checklist that is also reflected in the manuscript. The completed author checklist will also be part of the RPF.

5) Supplementary information:

- You can promote up to five figures to Expanded View (EV) Figures and Tables. These are collapsible/expandable online. If you wish to do so please cite EV Figures as 'Figure EV1, Figure EV2' etc... in the text and their respective legends should be included in the main text after the legends of regular figures.

- Appendix: For the figures that you do NOT wish to display as Expanded View figures, they should be bundled together with their legends in a single PDF file called *Appendix*, which should start with a short Table of Content including page numbers. Appendix figures should be referred to in the main text as: "Appendix Figure S1, Appendix Figure S2" etc. See detailed instructions regarding expanded view here:

- Please upload Table S1-S6, Table S8 and Table S10 as Datasets with the nomenclature Dataset EV1 with the legend in a separate tab of the .xls files. Alternatively, the legend for Datasets can be supplied as a separate text file (README) and zipped together with the Table/Dataset file.

- Table S7 and Table S9 can be part of the Appendix pdf or form EV tables.

6) Please note that all corresponding authors are required to supply an ORCID ID for their name upon submission of a revised manuscript (). Please find instructions on how to link your ORCID ID to your account in our manuscript tracking system in our Author guidelines

()

7) Before submitting your revision, primary datasets (and computer code, where appropriate) produced in this study need to be deposited in an appropriate public database (see).

The accession numbers and database should be listed in a formal "Data Availability " section (placed after Materials & Method), which you have already done.

8) We would also encourage you to include the source data for figure panels that show essential data. Numerical data should be provided as individual .xls or .csv files (including a tab describing the data). For blots or microscopy, uncropped images should be submitted (using a zip archive if multiple images need to be supplied for one panel). Additional information on source data and instruction on how to label the files are available .

9) Our journal encourages inclusion of *data citations in the reference list* to directly cite datasets that were re-used and obtained from public databases. Data citations in the article text are distinct from normal bibliographical citations and should directly link to the database records from which the data can be accessed. In the main text, data citations are formatted as follows: "Data ref: Smith et al, 2001" or "Data ref: NCBI Sequence Read Archive PRJNA342805, 2017". In the Reference list, data citations must be labeled with "[DATASET]". A data reference must provide the database

name, accession number/identifiers and a resolvable link to the landing page from which the data can be accessed at the end of the reference. Further instructions are available at .

10) Regarding data quantification:

- Please ensure to specify the name of the statistical test used to generate error bars and P values, the number (n) of independent experiments underlying each data point (not replicate measures of one sample), and the test used to calculate p-values in each figure legend. Discussion of statistical methodology can be reported in the materials and methods section, but figure legends should contain a basic description of n, P and the test applied.

IMPORTANT: Please note that error bars and statistical comparisons may only be applied to data obtained from at least three independent biological replicates. If the data rely on a smaller number of replicates, scatter blots showing individual data points are recommended.

- Graphs must include a description of the bars and the error bars (s.d., s.e.m.).

11) As part of the EMBO publication's Transparent Editorial Process, EMBO reports publishes online a Review Process File to accompany accepted manuscripts. This File will be published in conjunction with your paper and will include the referee reports, your point-by-point response and all pertinent correspondence relating to the manuscript.

I look forward to seeing a revised version of your manuscript when it is ready. Please let me know if you have questions or comments regarding the revision.

Yours sincerely

Martina Rembold, PhD
Editor
EMBO reports

Referee #1:

The authors of " A single-cell transcriptome atlas of the adult muscle precursors uncovers early events in fiber-type divergence in *Drosophila*" have made a bold and much needed analysis of the developing wing disc transcriptome at single cell resolution. The data this manuscript presents can be very useful in understanding wing disc cell type heterogeneity. However, some major concerns need to be addressed before publication.

Major comments:

1. In all third-instar larval preps and sequencing, it would be helpful to have the age of the animal

mentioned clearly. If possible, animal groups in all experiments should be age matched to within 6 hours. The differences between early and late 3rd instar wing discs might be significant. Though the data form apparently tight clusters, there may be an artefact of age differences being introduced in the data sets.

2. Gunage et al. 2014, distinguish the muscle lineage mononuclear cells on the 3rd instar wing disc into AMPs and their myoblast progeny. From text, like in the last paragraph on Page 3, there may arise confusion about the definitions of AMPs and myoblasts. Much of the text appears to exacerbate this confusion. The title of the study suggests the intention is to shed more light and clarity on these early stages in development. The authors need to not only make that distinction clear in the text but should, if possible, draw distinctions between the cell types from their data. That molecular basis of that distinction can be deduced from the data, as the authors demonstrate.

3. A very important conclusion that the writing suggests indirectly, is that Ama is an AMP marker. Key data here are the Ama>Gtrace shows that the expression of Ama is in the DFM precursor region of the wing disc (Fig 3H). In (Fig 3I), they show that Gtrace lineage expression in templates which is not persuasive. For reference, lineage GFP expression in Adult IFMs is clear. They further make the claim based on the 1151>Ama RNAi result, that it is essential for IFM formation. Ama maybe necessary for IFM development but experiments with clearer data to support its function in AMPs are perhaps necessary. We would interpret their results in this possible way:

a. ama-Gal4 is active in DFM precursors but not in DLM precursors. ama-GAL4 expression is limited to a few myoblasts near 16 hr apf DLM templates. ama-Gal4 is expressed strongly in developing templates some time post 16hrs apf and switches off in adults.

b. Because a single RNAi line has been used, that too with 1151-Gal4 which is expressed in all myoblasts, their conclusion that Ama is necessary in 3rd instar myoblasts (let alone AMPs) for IFM formation invites doubt.

To make this claim stronger, the authors must show:

a. Ama antibody staining in 3rd instar wing discs.

b. ama-Gal4> Ama RNAi (at least three validated RNAi lines targeting different exons). Also, they might make a more convincing statement about the effects of ama manipulation on all flight muscles simultaneously with improved methods of visualisation, such as MicroCT scanning (See Schoborg et al Development 2019, Chaturvedi et al Open Biology 2019).

c. ama>Gtrace induced between i) 0 to 20 hrs apf ii) 20-30 hrs apf iii) 30-50 hrs apf iv) 50hrs apf to 0 hour post eclosion to examine the ama-Gal4 activity and lineage at the end of each induction period. This should clarify if and when ama-Gal4 comes on in later stages of IFM development.

If they choose to remove data and statements regarding ama function, the manuscript could be for publication for the new datasets that they put forth: that can be of much value to the community.

4. The images in Fig 6G are too grainy. The mounting and imaging of the wing disc for this experiment needs at least to be comparable to Fig 5C.

Minor comments:

1. Fig 3E and G; 4B The wb>GFP, ArgK::GFP, E(spl)m3-HLH>GFP should be labelled on the image as opposed to labelling the images with GFP.

2. Fig 4D, F in a similar vein should be labeled edl-LacZ, Cgc25C-LacZ rather than B-gal.

3. The genotype of the wing discs in Fig 4E must be mentioned preferably on the image or in the legends.

In conclusion, this study is an important step in the right direction. With substantial revisions, experiments and closer examination and re-interpretation of the data, this manuscript will be acceptable for publication.

Referee #2:

This is an interesting manuscript by Zappia, Frolov and colleagues investigating the transcriptional diversity of *Drosophila* myoblasts at late larval stages using single cell sequencing techniques. The authors select parts of L3 wing discs as starting material and cell clustering can nicely distinguish the myoblasts that will form the future direct flight muscles (DFMs) from the ones that will form the indirect flight muscles (IFMs). This distinction is largely based on expression of 2 known markers, *vg* and *ct*, which are differential in the 2 classes. Interestingly, this work identifies a much larger number of differentially expressed genes between the 2 groups and hence can further subdivide the DFM myoblasts in 2 subgroups and the IFM myoblasts in 4 subgroups. Marker expression shows that these cells occupy distinct or partially overlapping areas on the wing disc depending on the clusters. These findings may implicate underappreciated functional distinctions of larval myoblasts.

In my opinion, these data are strong and provide a useful resource for future work. Minor revisions for clarity should be included.

However, the functional genetic data, in particular the functional analysis of a gene called *ama*, are currently weak and require major revisions before they can be published.

Major points.

1. I find it confusing that Figure 1G does not contain the known marker genes *vg* and *ct*, on basis of their expression DFM and IFM clusters were distinguished. Also other obvious genes, such as *twi* and *Mef2* should be included in 1G. They only come later or in the supplement, which makes the myoblast data very hard to follow throughout the paper.

2. A well known caveat of RNAi is off-targeting. Hence, phenotypic analysis needs to include a verification by a second independent RNAi construct or a rescue of the RNAi phenotype with an over-expressed cDNA or a genomic clone from a related species (*D. pseudoobscura*). Ideally, a genetic mutant is used to confirm the phenotype. Unfortunately, the potentially interesting phenotype upon *ama* knock-down is not confirmed with any of these. The authors test 2 RNAi lines, one is lethal, the other one has no phenotype. The authors assume that the second one does not work, but the first one does in a specific way. This needs confirmation.

The authors follow up on *ama*, as it is specifically expressed in the DFM myoblasts, however knock-down results in severe phenotypes in DFM and IFMs. How is this explained? Is *ama* expressed in IFM myoblasts at earlier stages? Where is the protein localized? Is it reduced upon knock-down? NB: the data on *chinmo* look interesting, several hairpins are lethal upon *Mef2* induced knock-down, but little follow up is in the paper.

3. Figure 6E investigates marker expression in *ama* RNAi myoblasts and concludes that many markers are lost upon knock-down. However, myoblasts number is severely affected in the knock-down. Would it not be important to normalize the amount of myoblasts that express the markers to the total amount of myoblasts rather than to the amount of epithelial cells as currently done? The text on page 21 and 22 does not match to Figures 6D and F.

Minor points.

1. Please explain how the number of 'Average Expression' in Figure 1G and following Figures has been generated. Are these normalized read counts? Log scale? What was used for normalization. This should be briefly stated in the text or legend.
2. The text reports 24 clusters, however Figure 1 has 26. The labeling of the clusters is highly confusing. Why not using IFM1-6 and epithelial 1-12, plus the other special ones? The reader only understands some, by far not all of the names at a rather late stage of the paper.
3. How do the epithelial clusters in Figure 2 compare to recent work by the Boutros and Teleman labs (Bageritz Nat Methods 2019)?
4. Only half of the IFMs use a template mechanism to form at pupal stages (the dorsal longitudinal flight muscles, DLMs), the dorsoventral flight muscles (DVMs) use the normal founder cell mechanism, as the DFMs do. Thus, the statement and line of argumentation for founder cell specific genes in DFM myoblasts on page 13 are incorrect.
5. What is the prominent expression pattern of wb-GAL4 driven GFP in Figure 3E?
6. Cg25C appears high (red) in IFM1, but authors use it to locate IFM2 in wing discs, page 18 Fig. 4F.
7. Schnorrer et al. 2010 did a very similar functional approach using Mef2-GAL4 to knock-down genes with RNAi. Knock-down of SPARC was reported lethal and Argk was reported as pharate lethal, hence identical to reported here. This could be mentioned.
8. Figure 5D does not seem to be referenced in the text.
9. The RNAi resource papers, Dietzl et al. 2007, Ni... Perrimon Nat Meth 2011 should be cited, as many hairpins were used.

Referee #3:

The authors in the present manuscript set out to better understand the AMP cell population associated the developing flight muscle and how this population splits and gives rise to direct and indirect flight muscle.

Using a single cell transcriptomics approach, the tissue is extracted, cells are sequenced and major populations of epithelial cells and AMPs are identified.

Further clustering identifies smaller subpopulation of the epithelial, 2 populations of a tracheal lineage, and several populations of the direct and of the indirect flight muscle.

Studying the gene expression differences in these diverse populations, evidence is presented that many populations are spatially restricted. New Marker genes are tested and confirmed. This allows the authors not only to distinguish between the direct and indirect flight muscle lineage, but smaller subpopulations within.

Furthermore, the authors test differentially expressed genes for functionality in flight muscle development. They ultimately focus on the most promising candidate - ama - and find that upon downregulation by RNAi, the AMP population is severely depleted.

It should be noted, that the set-up is rather elegant in that the "WT" strains used were the driver

strain and a driver crossed to a mock-RNAi (mCherry) strain. While this will not be a perfect match to, for example, the RNAi cross against for Ama down regulation, the clustering similarity is very convincing for the generalizability of the results.

The highlight is the single cell comparison of cell types in wt and Ama knockdown disks. While the argument of proliferative deficiency versus increased cell death may not be entirely convincing, the effect of severe AMP depletion is crystal clear.

Major questions:

- It is indicated that cells that were unevenly represented between replicates were removed. Please specify (a) what cells these might be, and (b) could you somehow indicate in FigS1 (I cannot see them comparing S1B with 1C)
- Please provide Violin plots indicating gene detection and UMI depth for replicates in S1 (you might also want to show the global average of retained cells. I assume AMPs and epithelial cells would not differ much)
- why are cut and vestigial not part of 1G?
- I cannot clearly read the genes in 1D, but why is cut specific to / much higher in one DFM population. It was stated in the introduction that cut should be on in both AMP populations, DFM AND IFM. vg makes more sense but also seems to experience more regulation than expected among the IFM populations. Confusingly, in the results section, the authors contradict themselves compared to the intro by stating that cut is DFM-specific.
- I find it more than surprising that not a single non-coding RNA (CR's) is differentially expressed anywhere. Were they filtered?
- are the two tracheal populations you identify spatially separated?
- Figures 2 and S2 should be supported by staining for gene expression, especially for new marker genes. I find the discussion about cell cluster location intriguing, but would be more convinced if supported by hard evidence to test their predictions. For example, Expression of dpp in Epi_16 is argued to mean that these cells belong in the anterior hinge. However, one might note that while the cells that express it do so highly, only ~50ish percent of the cells in Epi_16 are found to express it. This would be more easily explained by drop-outs if the expression levels were low.
- Is there any way to provide higher quality expression images? I am not too familiar with the challenges of staining a wing imaginal disk, but it would seem possible to better support the spatial predictions in Figure 3 (a major result) with better imaging. Case in point: aside from image sizes being all over the place, one antibody that bothers me in particular is Ct. The authors use it to identify the DFM population, but the images do not let the reader make that distinction clearly. Some may be due to background, more may be due to low level cut expression in the IFM populations (supported by 3B). But I cannot help but wonder if RBNA-ISH might do a better job here. Furthermore, their detection in single cells is RNA, so ISH would better represent the data they describe. Also, in some panels I see the nuclear signal I expect from cut, but not in others.
- another problem that may just be an imaging problem is that I cannot see in Fig3H what they say I should see: "GFP was broadly expressed in both IFM and DFM precursors, thus indicating that Ama was expressed in IFM precursors earlier in development (Figure 3H)". First, In the downloaded images I cannot see GFP broadly in the IFM. Second, Given the strength of the GFP lineage trace in adults in 3I, I am not convinced that there is real GFP signal in the IFM at 16h APF.
- no statistical tests are indicated for the screen results (Figure 5). More generally, I think the screening section should be described better. It is not clear to me from the data presented which lines were tested not only with the Mef2 driver, but also with the 1151-Gal4 driver. Furthermore, no results for the 1151 driver are presented in the figure as far as I can see. Statements such as "the few Mef2>stg-RNAi animals that survived to adulthood were flightless" seem to go against Figure 5A that shows 50% make it to adults, and 5B shows that just about 75% land elsewhere than the

bottom of the tube. Similarly, calling the RNAi animals for the elongation factor 'flightless' seems like an overstatement.

- the authors propose due to the presence of E(Spl) genes that 3 IFM populations are notch responsive. I think this is interesting enough and should be shown by, for example, N[ICD] stains. This would be especially powerful if you could co-visualize it with a pan-IFM marker and maybe Con or stg versus DNA-J1 or Cg25c
- The authors generally presume that their clusters have distinct spatial identities. Why? Why would clusters not represent distinct cell type identities that spatially intermingle?
- the argument that subpopulations may represent different differentiation states is interesting. It should be feasible to show this along pseudotime.
- I am intrigued by the fact that Ama downregulation depletes all AMP populations (DFM and IFM). This supports the TRACE experiments I criticized earlier, but more discussion is warranted.

Minor points:

- I do not understand what figure 1F does by itself. It should be part of 1E as visual support for the identity of the major separation, but it is not a novel result that deserves its own panel.
- The authors use GFP reporter lines to confirm their expression prediction (e.g. SPARC) - it seems to be work and it is believable, though RNA-ISH would have been more direct and a better readout. Was there a reason for this choice?
- when indicating average expression in dot plot color, can you give units? I assume this is some highly normalized value, taking into account filtered UMI proportion per cell and per-cell UMI depth, but how can you get negative values? (I also assume this is average expression in cells where the gene was detected, not all cells in the cluster (as this is the Seurat default))
- I believe the legend in 3A obscures data
- I have not seen this flight test before.... how do you prevent that flies get stuck on the paper while you transfer flies, or while they fly up? in other words, why is their place of demise a landing event?

General notes on style and length:

LENGTH:

- The paper is written largely clearly, but would benefit from writing in a much more concise manner. For example, information such as AFPs being precursors to DFMs and IFMs is given multiple times (abstract, intro, figure legend...) and instances like that could be eliminated. Especially the figure legends should be shortened.
- The current section on spatial assignments on epidermal populations - while interesting - is repetitive and hard to read. For the general reader, this section might be shortened and supported by a table. And for the expert, a supplementary note with all the details might be supplied.
- There are several grammatical problems (articles, etc.) that should be addressed, but nothing that the type editors won't catch.
- "access" versus "assess"
- To frame this investigation in terms of uncovering if ct and vg are the only two differentially expressed genes seems unreasonable. I doubt anybody would have put money on that, including the authors. Furthermore, per the author's description, it is only vestigial that would be differentially expressed between the populations... cut (they say) is expressed in both.

Referee #1:

The authors of "A single-cell transcriptome atlas of the adult muscle precursors uncovers early events in fiber-type divergence in Drosophila" have made a bold and much needed analysis of the developing wing disc transcriptome at single cell resolution. The data this manuscript presents can be very useful in understanding wing disc cell type heterogeneity. However, some major concerns need to be addressed before publication.

Response: We are pleased that the reviewer appreciates the need of this study in the field, and we thank him/her for helpful comments on how to improve the manuscript. Our revised manuscript addresses all comments raised by the reviewer. Additionally, we have strengthened the cell atlas by including the visualization of six new myoblast markers in the wing discs (Figures 3E, 3H, 4B, 4C, 5E and Figure EV4F). Also, the loss of function screen was expanded to include eleven additional markers (Figure 6A-D). More importantly, we inferred the trajectory of IFM myoblast lineage using an *in silico* approach (Figure 4E-G) and we genetically traced the lineage of AMP cells using the G-TRACE tool to confirm our findings (Figure 4H-I). Finally, we validated the specificity of the *UAS-Ama-RNAi* line by rescuing the lethality phenotype using an *Ama*-cDNA construct in the background of *1151>Ama-RNAi* (Figure 7A). Changes are highlighted in the revised text.

Major comments:

1. *In all third-instar larval preps and sequencing, it would be helpful to have the age of the animal mentioned clearly. If possible, animal groups in all experiments should be age matched to within 6 hours. The differences between early and late 3rd instar wing discs might be significant. Though the data form apparently tight clusters, there may be an artefact of age differences being introduced in the data sets.*

Response: The reviewer is raising a great point here. We are participating in the Fly Cell Atlas meetings and are aware of recent reports about the differences in gene expression pattern associated with age. However, our experimental design does not provide the resolution needed to fully address this effect and, at this point, we cannot account for age-specific changes in gene expression. The dataset we report here integrates eight replicates of two wild type genotypes to account for genetic background variability (Figure EV1A-B). These samples were collected at wandering third instar larval stage, which is roughly 110-135 h AEL at 25 °C. This information was added in the sections Results (page 6) and Materials & Methods (page 35).

2. *Gunage et al. 2014, distinguish the muscle lineage mononuclear cells on the 3rd instar wing disc into AMPs and their myoblast progeny. From text, like in the last paragraph on Page3, there may arise confusion about the definitions of AMPs and myoblasts. Much of the text appears to exacerbate this confusion. The title of the study suggests the intention is to shed more light and clarity on these early stages in development. The authors need to not only make that distinction clear in the text but should, if possible, draw distinctions between the cell types from their data. That molecular basis of that distinction can be deduced from the data, as the authors demonstrate.*

Response: We apologize for the confusion. Text was thoroughly revised and clarified to accurately reflect the distinction between these two types of cells (pages 3, 18, 20, 28). Accordingly, we used the term myoblast, which is generally used, to refer to all cells in the ad epithelial layer that form the flight muscles. The labels of clusters were reassigned following the suggestion of Reviewer 2 (DFM_1-2 and IFM_1-5). Briefly, clusters IFM_1-2 were reassigned as AMPs, the adult muscle precursor cells, and the clusters IFM_4-5 as differentiating myoblasts. Also, our data suggest that the cluster IFM_3 is transitioning from AMPs to differentiating myoblasts. The IFM myoblast cell lineage was confirmed by *in silico* and *in vivo* approaches: cells were traced using the genetic tool G-TRACE (Figure 4H-I, page 18) and pseudotime was inferred using Slingshot (Figure 4E-G, page 18).

The molecular basis that make this distinction is Notch signaling pathway, which is in agreement with the work by (Gunage *et al*, 2014). The AMPs are Notch-positive cells, whereas differentiating myoblasts no longer show active Notch in their transcriptomes. The Notch signature is apparent in the clusters IFM-1_2 and DFM_1.

3. A very important conclusion that the writing suggests indirectly, is that *Ama* is an AMP marker. Key data here are the *Ama>Gtrace* shows that the expression of *Ama* is in the DFM precursor region of the wing disc (Fig3H). In (Fig3I), they show that *Gtrace* lineage expression in templates which is not persuasive. For reference, lineage GFP expression in Adult IFMs is clear. They further make the claim based on the 1151>*Ama* RNAi result, that it is essential for IFM formation. *Ama* maybe necessary for IFM development but experiments with clearer data to support its function in AMPs are perhaps

necessary. We would interpret their results in this possible way:

a. *ama-Gal4* is active in DFM precursors but not in DLM precursors. *ama-GAL4* expression is limited to a few myoblasts near 16 hr apf DLM templates. *ama-Gal4* is expressed strongly in developing templates some time post 16hrs apf and switches off in adults.

Response: We carried out G-TRACE experiments at a later time points during pupal development as the reviewer requested. The DLM (IFM) were stained at 48h and 72h APF in *Ama>G-TRACE* animals to determine whether *Ama* is expressed in DLM during pupa development. We found that DLM were GFP-positive and RFP-negative (Figure R1), thus suggesting that *Ama* was expressed at earlier stages of development (*GAL4* lineage, green), but likely not during pupal development (active *GAL4*, red). Since it is not clear when *Ama* is

expressed in pupa development, we decided to remove data for DLM staining at 16h APF from the revised manuscript and restated our conclusions (please see below).

b. Because a single RNAi line has been used, that too with 1151-Gal4 which is expressed in all myoblasts, their conclusion that *Ama* is necessary in 3rd instar myoblasts (let alone AMPs) for IFM formation invites doubt.

To make this claim stronger, the authors must show:

a. *Ama* antibody staining in 3rd instar wing discs.

Response: We have tested anti-*Ama* serum (Seeger *et al*, 1988) by IF, but the results were inconclusive. We have performed *fluorescent in situ hybridization* for *Ama* mRNA expression in wandering third instar larval wing discs (Figure 3 J-K), and provide images for *Ama>GFP* in the wing discs (Figure 3I). We showed that the pattern revealed by FISH largely overlapped with the pattern of *Ama>GFP*. Our data led us to conclude that *Ama* is expressed in DFM myoblasts and in a small subset of IFM myoblasts (yellow asterisk and arrowhead, respectively, Figure 3I-K, page 15). This subset of cells appears to overlap with the localization of IFM_1-2 cluster cells (Figure 4B-D, page 18), which is consistent with low expression of *Ama* in the cluster IFM_1 as revealed by scRNA-seq (Figure 3B). Since the clusters IFM_1-2 correspond to AMPs and these cells will contribute to expand the population of IFM myoblasts (*GFP*, *GAL4* lineage, Figure 4H-I, page 18), the loss of *Ama* in some cells of the cluster IFM_1 may alter the pool of AMPs, and subsequently impairs the massive expansion of IFM myoblasts (Figure 7D,H) and the formation of IFM (Figure 8B,D).

Also, please see reply to comment 2 for Referee 2 and the last mayor comment for Referee 3.

b. ama-Gal4> Ama RNAi (at least three validated RNAi lines targeting different exons). Also, they might make a more convincing statement about the effects of ama manipulation on all flight muscles simultaneously with improved methods of visualisation, such as MicroCT scanning (See Schoborg et al Development 2019, Chaturvedi et al Open Biology 2019).

Response: We tested an additional *UAS-Ama* RNAi line targeting a different region of *Ama* gene. We added *UAS-Dicer2* in the background to enhance the RNAi effect. We found a significant decrease in animal viability upon knockdown of *Ama* using the pan-muscular driver *Mef2-GAL4* (Figure 6G, page 23). Thus, our data confirm that *Ama* is required in muscles. Moreover, in order to validate the *UAS-Ama-RNAi* used here, the lethality phenotype was partially rescued by overexpressing the *UAS-Ama* (cDNA) construct (Figure 7A, page 23), thus demonstrating that *UAS-Ama-RNAi* is targeting specifically *Ama* gene.

We agree that MicroCT scanning is a great technique to examine muscle morphology and may help to reveal what we missed by IF. However, this will unlikely change the overall conclusion because *Ama*-depleted thoraces largely lack the flight muscles. This is due to a severe loss of myoblasts at earlier stages of development (Figure 7D).

c. ama>Gtrace induced between i) 0 to 20 hrs apf ii) 20-30 hrs apf iii) 30-50 hrs apf iv) 50hrs apf to 0 hour post eclosion to examine the ama-Gal4 activity and lineage at the end of each induction period. This should clarify if and when ama-Gal4 comes on in later stages of IFM development.

If they choose to remove data and statements regarding ama function, the manuscript could be for publication for the new datasets that they put forth: that can be of much value to the community.

Response: The G-TRACE tool that we are using here is not an inducible system (Evans *et al*, 2009), thus the suggested experiment is not feasible.

However, we have analyzed both real-time and cell lineage expression of *Ama* using three methods: *UAS-GFP* reporter, *G-TRACE* lineage tracing and FISH in the wing discs. New images were added to Figure 3I-K and Figure EV3C (page 15). Overall, we conclude that *Ama* is expressed in DFM myoblasts and in a small subset of IFM myoblasts in larval wing disc (yellow arrowhead, Figure 3I-K, page 15).

Experiments done at later stages of IFM development, including 48h and 72h APF (Figure R1), suggest that *Ama* was expressed at earlier stages of development (*GAL4* lineage, green), but likely not during pupal development (active *GAL4*, red). We removed IF done for DLM at 16h APF from the revised manuscript. Please see reply to comment 3.a

4. The images in Fig6G are too grainy. The mounting and imaging of the wing disc for this experiment needs at least to be comparable to Fig 5C.

Response: Images were replaced to meet required resolution. Please see now Figure 7I.

Minor comments:

1. Fig 3E and G; 4B The wb>GFP, ArgK::GFP, E(spl)m3-HLH>GFP should be labelled on the image as opposed to labelling the images with GFP.

Response: Labels were changed.

2. Fig 4D, F in a similar vein should be labeled edl-LacZ, Cgc25C-LacZ rather than B-gal.

Response: Labels were changed.

3. The genotype of the wing discs in Fig 4E must be mentioned preferably on the image or in the legends.

Response: Genotype was added

In conclusion, this study is an important step in the right direction. With substantial revisions, experiments and closer examination and re-interpretation of the data, this manuscript will be acceptable for publication.

Referee #2:

This is an interesting manuscript by Zappia, Frolov and colleagues investigating the transcriptional diversity of Drosophila myoblasts at late larval stages using single cell sequencing techniques. The authors select parts of L3 wing discs as starting material and cell clustering can nicely distinguish the myoblasts that will form the future direct flight muscles (DFMs) from the ones that will form the indirect flight muscles (IFMs). This distinction is largely based on expression of 2 known markers, vg and ct, which are differential in the 2 classes. Interestingly, this work identifies a much larger number of differentially expressed genes between the 2 groups and hence can further subdivide the DFM myoblasts in 2 subgroups and the IFM myoblasts in 4 subgroups. Marker expression shows that these cells occupy distinct or partially overlapping areas on the wing disc depending on the clusters. These findings may implicate underappreciated functional distinctions of larval myoblasts.

In my opinion, these data are strong and provide a useful resource for future work. Minor revisions for clarity should be included.

However, the functional genetic data, in particular the functional analysis of a gene called ama, are currently weak and require major revisions before they can be published.

Response: We thank the reviewer for appreciating the value of our work, and providing thoughtful comments and suggestions. We added data to strengthen the functional role of Ama in muscle. We further characterized Ama expression during development, in particular in the wing discs of third instar larvae using both FISH and G-TRACE techniques (Figure 3I-K and Figure EV3C). We validated the specificity of UAS-Ama-RNAi line using two approaches: (1) a 2nd independent RNAi line with UAS-dicer2 in the background (Figure 6G), and (2) an UAS-Ama construct to partially rescue the lethality of 1151>Ama-RNAi (Figure 7A). Finally, we extended the functional screen for the markers identified in the myoblast atlas (Figure 6A-D), and deciphered the temporal progression of the myoblast clusters by combining a genetic cell lineage tracing technique with pseudotime inference (Figure 4E-I). Changes are highlighted in the revised text.

Major points.

1. I find it confusing that Figure 1G does not contain the known marker genes vg and ct, on basis of their expression DFM and IFM clusters were distinguished. Also other obvious genes, such as twi and Mef2 should be included in 1G. They only come later or in the supplement, which makes the myoblast data very hard to follow throughout the paper.

Response: The expression of vg, ct, twi and Mef2 were added to dot plot in Figure 1G (page 7). As expected vg expression is also found in epithelial clusters and ct in tracheal cells as well as in SOPs (Pitsouli & Perrimon, 2010; Blochlinger *et al*, 1990; Williams *et al*, 1991). The percentage of cells that captured Mef2 mRNA is fairly low. This is most likely due to the limitation of the Drop-seq technique that is not as sensitive as the commercial 10x Genomics platform.

2. A well known caveat of RNAi is off-targeting. Hence, phenotypic analysis needs to include a verification by a second independent RNAi construct or a rescue of the RNAi phenotype with an overexpressed cDNA or a genomic clone from a related species (D. pseudoobscura). Ideally, a genetic mutant is used to confirm the phenotype. Unfortunately, the potentially interesting phenotype upon ama knock-down is not confirmed with any of these. The authors test 2 RNAi lines, one is lethal, the other one has no phenotype. The authors assume that the second one does not work, but the first one does in a specific way. This needs confirmation.

Response: The reviewer raised an important point. Unfortunately, Ama mutants are not publicly available. However, we confirmed the results of Ama depletion by rescuing the lethality with an Ama transgene and by using another independent RNAi line. The details of these experiments are the following.

First, We generated *UAS-Ama* transgene to overexpress the *Ama* cDNA in the background of the first *UAS-Ama RNAi* [TRiP. HMS00297] line. The *UAS-Ama* partially rescued the lethality phenotype of *1151>UAS-Ama RNAi* [TRiP. HMS00297]. New data are included in Figure 7A (page 23). Secondly, we added *UAS-Dicer2*, which enhances the RNAi machinery, to the background of the second *UAS-Ama[GD12733] RNAi* line, the one that did not initially show a phenotype. We found a significant reduction in viability upon *Ama* depletion with *Mef2-GAL4* (Figure 6G, page 23). The viability test of *UAS-Ama[GD12733] RNAi* line (without *UAS-Dicer2*) that was in initial submission was removed from the manuscript to avoid any confusion. Thus, new findings confirmed that *Ama* is required for muscle development.

The authors follow up on ama, as it is specifically expressed in the DFM myoblasts, however knockdown results in severe phenotypes in DFM and IFMs. How is this explained? Is ama expressed in IFM myoblasts at earlier stages? Where is the protein localized? Is it reduced upon knock-down?
NB: the data on chinmo look interesting, several hairpins are lethal upon Mef2 induced knock-down, but little follow up is in the paper.

Response: We have tested anti-*Ama* serum (Seeger *et al*, 1988), but the results were inconclusive. Thus, we could not analyze localization of *Ama* protein. However, using a variety of techniques, including *Ama-GAL4* reporter, *FISH* and cell lineage tracing with G-TRACE, we conclude that *Ama* is expressed in DFM myoblasts and in a small subset of IFM myoblasts. Data were reorganized and included in Figure 3I-J (page 15). Also, we confirmed that *Ama* is expressed in some IFM myoblasts at earlier stages as indicated by G-TRACE lineage tracing (GFP, Figure 3K and Figure EV3C, page 15). This region appears to overlap with the region where IFM_1-2 cluster cells are located, which is consistent with *Ama* being detected in IFM_1 (Figure 3B). Findings using G-TRACE approach confirmed that the cells in clusters IFM_1-2 contribute to the expansion of IFM myoblasts (Figure 4H-I, page 18). Thus, the loss of *Ama* in some cells of cluster IFM_1 likely impaired the pool of IFM myoblasts. This may help to explain why IFM myoblasts are severely affected upon *Ama*-depletion (Figure 7D), which consequently impairs the formation of IFM (Figure 8B,D). This is additionally reflected by the severe loss of both IFM and DFM myoblasts in *Ama-RNAi* discs at early third instar larva (Figure 7H), most likely through the inhibition of myoblast expansion (Figure EV5D-E).

Given that *Ama* is also secreted (Fremion *et al*, 2000) we cannot exclude the possibility of a non-cell autonomous effect to explain the severe phenotype in the formation of IFM.

Based on scRNA-seq data, *Ama* expression was reduced in myoblasts as shown in Figure 7E. Thus, the *UAS-Ama-RNAi* targets *Ama* gene and reduces its expression.

We agree that *chinmo* is a very attractive candidate. However, we initially pursued the analysis of *Ama* function in muscle and therefore we included it in this manuscript.

3. Figure 6E investigates marker expression in *ama RNAi* myoblasts and concludes that many markers are lost upon knock-down. However, myoblasts number is severely affected in the knock-down. Would it not be important to normalize the amount of myoblasts that express the markers to the total amount of myoblasts rather than to the amount of epithelial cells as currently done?

The text on page 21 and 22 does not match to Figures 6D and F.

Response: One of the major advantages of scRNA-seq is that it allows detection of rare cell types. This is because the level of gene expression in the individual cells is not affected by the number of cells, as the transcriptome of each cell is profiled by an individual and uniquely barcoded single cell library. The dot plot in Figure 7E of the revised manuscript shows the expression of gene markers in *Ama-RNAi* myoblast and in wild type. We used Seurat to normalize data and compare these two genotypes. This method “employs a global-scaling normalization method “LogNormalize” that normalizes the feature expression measurements for each cell by the total expression, multiplies this by a scale factor (10,000 by default), and log-transforms the result” (Stuart *et al*, 2019). Therefore, a reduction in the number of cells will not

affect the interpretation of data because the expression value is normalized to each cell. The power of scRNA-seq is that it takes the information from each individual cell disregarding the number of cells left. In sum, low number of cells will not show a bias in the data. Therefore, the reduced expression of gene markers in *Ama-RNAi* is not the result of low number of myoblasts.

In Figure 7F we projected the *Ama-RNAi* dataset onto the reference dataset. Each individual *Ama-RNAi* cell was classified based on the reference cell atlas. This analysis returned a matrix with predicted cell clusters along with prediction scores for each *Ama-RNAi* cell. The graph is showing how many *Ama-RNAi* cells were classified as DFM_1, DFM_2, IFM_1, IFM_2, IFM_3, IFM_4 or IFM_5 based on gene expression similarity with the reference dataset. Because there is a drastic bias in the number of myoblasts in *Ama-RNAi* compared to control, we decided to use the total amount of cells to normalize the number of cells projected into each reference cluster.

I hope this answers the reviewer's question. The information regarding normalization was added in the section Materials & Methods (page 37).

The reference to the Figure 7 was corrected.

Minor points.

1. Please explain how the number of 'Average Expression' in Figure 1G and following Figures has been generated. Are these normalized read counts? Log scale? What was used for normalization. This should be briefly stated in the text or legend.

Response: Dot plots use the expression value of each gene that is first normalized and then scaled, as indicated in Seurat platform (Stuart *et al*, 2019).

First, the global-scaling normalization method normalizes gene expression for each cell by the total expression, then multiplies this by a scale factor (10,000), and log-transforms the result. Next, linear transformation (or scaling) is applied. The expression of each gene is shifted, so that the mean expression across cells is 0, and is scaled, so that the variance across cells is 1. This step gives equal weight in downstream analyses, so that highly-expressed genes do not dominate.

In summary, cells with a value > 0 represent cells with expression above the population mean (a value of 1 would represent cells with expression 1SD away from the population mean).

This information was added in the section Materials & Methods (page 37), and legend of Figure 1 showing the first dot plot (pages 56-57).

2. The text reports 24 clusters, however Figure 1 has 26. The labeling of the clusters is highly confusing. Why not using IFM1-6 and epithelial 1-12, plus the other special ones? The reader only understands some, by far not all of the names at a rather late stage of the paper.

Response: We apologize for the confusion. Text was revised (page 13), and labels of clusters were changed to DFM_1-2 and IFM_1-5. However, we would like to keep the labels of epithelial clusters and others to better reflect the subtype of cells. We hope the reviewer would agree.

3. How do the epithelial clusters in Figure 2 compare to recent work by the Boutros and Teleman labs (Bageritz *Nat Methods* 2019)?

Response: In (Bageritz *et al*, 2019), the authors sequenced 4,198 cells with Drop-seq and 10xGenomics, and focused only on the wing disc proper. In contrast, our work recovered and clustered 11,527 cells with Drop-seq. Thus, besides the wing disc proper we were able to analyze diverse cell types in the imaginal wing disc, including peripodial membrane, external sensory organs, myoblasts and tracheal cells, though our work did not include the wing pouch.

Although the gene expression map in Bageritz *et al* is based on gene expression correlation rather than clustering based on gene expression similarity as we did, both studies found same new markers for wing

disc proper: CR44334 and kank/ CG10249. However, Bageritz *et al* lacks markers for peripodial membrane, myoblasts, tracheal cells, and SOPs.

Overall, the main distinction of our atlas is a larger number of cells analyzed that results in 1.8x cell coverage for myoblasts and reveals distinct states of AMPs differentiation.

4. *Only half of the IFMs use a template mechanism to form at pupal stages (the dorsal longitudinal flight muscles, DLMs), the dorsoventral flight muscles (DVMs) use the normal founder cell mechanism, as the DFM do. Thus, the statement and line of argumentation for founder cell specific genes in DFM myoblasts on page 13 are incorrect.*

Response: We appreciate the comment. Text was adjusted on page 14.

5. *What is the prominent expression pattern of wb-GAL4 driven GFP in Figure 3E?*

Response: Since this enhancer trap transgene showed high expression in air sac, and our dataset did not indicate *wb* expression in other clusters, we decided to remove images from manuscript and replace it with more reliable markers. Revised manuscript includes the localization of 6 new markers for myoblast clusters, including Ten-a, nkd, E(spl)m6-BFM, E(spl)m7-HLH, E(spl)mbeta-HLH and Vkg (Figures 3E, 3H, 4B, 4C, 5E, and Figure EV4F, pages 14, 17, 20). Also, we used two independent reporters to validate spatial localization over the discs for vkg, Cg25C and edl (Appendix Figure S2, pages 19-20).

6. *Cg25C appears high (red) in IFM1, but authors use it to locate IFM2 in wing discs, page 18 Fig. 4F.*

Response: We appreciate the comment. We realized there was an issue with the dot plot in Figure 4A. It has been corrected now.

7. *Schnorrer et al. 2010 did a very similar functional approach using Mef2-GAL4 to knock-down genes with RNAi. Knock-down of SPARC was reported lethal and Argk was reported as pharate lethal, hence identical to reported here. This could be mentioned.*

Response: We apologize for missing this. References were added to Results (pages 21-22) and kept in Discussion (page 29).

8. *Figure 5D does not seem to be referenced in the text.*

Response: Figure reference is now added to the revised text as Figure 6F (page 22).

9. *The RNAi resource papers, Dietzl et al. 2007, Ni... Perrimon Nat Meth 2011 should be cited, as many hairpins were used.*

Response: References were added to the revised text (pages 21, 31).

Referee #3:

The authors in the present manuscript set out to better understand the AMP cell population associated the developing flight muscle and how this population splits and gives rise to direct and indirect flight muscle.

Using a single cell transcriptomics approach, the tissue is extracted, cells are sequenced and major populations of epithelial cells and AMPs are identified.

Further clustering identifies smaller subpopulation of the epithelial, 2 populations of a tracheal lineage, and several populations of the direct and of the indirect flight muscle.

Studying the gene expression differences in these diverse populations, evidence is presented that many populations are spatially restricted. New Marker genes are tested and confirmed. This allows the authors not only to distinguish between the direct and indirect flight muscle lineage, but smaller subpopulations within.

Furthermore, the authors test differentially expressed genes for functionality in flight muscle development. They ultimately focus on the most promising candidate - ama - and find that upon

downregulation by RNAi, the AMP population is severely depleted. It should be noted, that the set-up is rather elegant in that the "WT" strains used were the driver strain and a driver crossed to a mock-RNAi (mCherry) strain. While this will not be a perfect match to, for example, the RNAi cross against for *Ama* down regulation, the clustering similarity is very convincing for the generalizability of the results. The highlight is the single cell comparison of cell types in wt and *Ama* knockdown disks. While the argument of proliferative deficiency versus increased cell death may not be entirely convincing, the effect of severe AMP depletion is crystal clear.

Response:

We thank the reviewer for his/her interest in our work and the constructive remarks and suggestions. We hope that the reviewer agrees that new experiments included in the revision have strengthened the manuscript. All comments and suggestions were addressed below. Briefly, the localization of both epithelial and myoblast markers were examined in the wing discs to validate our findings (Figures 2F, 3E, 3H, 4B, 4C, 5E and Figure EV4F). We uncovered the temporal progression of the distinct cellular states by both *in silico* and *in vivo* approaches. Cell lineages were genetically traced and pseudotime inferred (Figure 4E-I). Changes are highlighted in the revised text.

Major questions:

- It is indicated that cells that were unevenly represented between replicates were removed. Please specify (a) what cells these might be, and (b) could you somehow indicate in FigS1 (I cannot see them comparing S1B with 1C)

Response: We used unbiased stringent criteria to select only good quality cells. After overclustering, we removed clusters that were specific to one or two replicates because these were most likely due to either technical or dissection issues. We could not tell what cells these might be though. Cell clustering prior to removal of unevenly represented clusters across replicates was included in revised manuscript in Appendix Figure S1A (page 6), along with the percentage of cells per replicate in each cluster in Appendix Figure S1B. Clusters removed from downstream analysis are indicated with red arrows. Moreover, we identified and removed clusters expressing markers of either cellular stress or dying cells, such as heat shock protein, rRNA and high mitochondrial content. Also, we monitored the number of transcripts per cluster and whether known epithelial and myoblast markers were co-expressed in specific clusters to find potential multipliers. Detailed information was added in Materials & Methods (page 38). Figure EV1A-B shows cell clustering once low-quality cells were removed.

- Please provide Violin plots indicating gene detection and UMI depth for replicates in S1 (you might also want to show the global average of retained cells. I assume AMPs and epithelial cells would not differ much)

Response: Plots for *nfeature_RNA* and *nCount_RNA* are included in Figure EV1C-D (page 6).

- why are *cut* and *vestigial* not part of 1G?

Response: We apologize we missed these important markers. The expression of *cut* and *vestigial* were added to Figure 1G in revised manuscript (page 7).

- I cannot clearly read the genes in 1D, but why is *cut* specific to / much higher in one DFM population. It was stated in the introduction that *cut* should be on in both AMP populations, DFM AND IFM. *vg* makes more sense but also seems to experience more regulation than expected among the IFM populations. Confusingly, in the results section, the authors contradict themselves compared to the intro by stating that *cut* is DFM-specific.

Response: *ct* has been previously identified has a gold standard marker to distinguish DFM myoblasts from IFM myoblasts as shown in Figure 1, 6 and 7 from (Sudarsan *et al*, 2001) and in Figure 3C in our revised manuscript. However, there is a clear limitation in the visualization of *ct* by immunofluorescence

since there is only a change in gene expression, and not a switch ON/OFF. So, there is an urgent need to find new markers to unambiguously label these two types of myoblasts.

In order to confidently classify myoblast types, the data of both markers *ct* and *vg* were integrated. The marker *ct* was highly expressed in DFM_1-2 clusters compared to IFM_1-5 clusters (Figure 3B). The cluster DFM_2 showed reduced levels of *ct* expression compared to DFM_1. However, these levels were still higher than IFM_1-5 clusters. Because data in dot plot is scaled (see below), *ct* expression in IFM was actually below the average expression, and therefore showed negative values, as opposed to dot plot in Figure 1G, which included the whole dataset. All IFM_1-5 clusters showed some levels of expression of *ct* compared to epithelial cell clusters (Figure 1G).

Overall, *vg* is expressed in IFM_1-5 clusters but not in DFM_1-2 clusters (Figure 1G and Figure 3B). The expression of *vg* is higher in IFM_2 compared to the other IFM clusters, which has not been previously reported by IF. Thus, there is likely some Wingless signaling regulation across the clusters of IFM myoblasts. This is consistent with previous reports (Sudarsan *et al*, 2001; Gunage *et al*, 2014).

Text was revised to clarify this and remove any contradictions (pages 4, 13).

- I find it more than surprising that not a single non-coding RNA (CR's) is differentially expressed anywhere. Were they filtered?

Response: Non-coding RNAs were not filtered. These can be found in Figure 2E. CR44334 and CR44811 are markers for cluster Epi_9, and pncr002;3R is a marker for cluster ES_3. Overall, most non-coding genes did not score as top 3 to 5 markers, but can be found on the list of biomarkers in Dataset EV2, Dataset EV3, and Dataset EV4. We do not know why the number of non-coding RNAs is low in our dataset though.

- are the two tracheal populations you identify spatially separated?

Response: Yes, these are two spatially separated tracheal cell types associated with the wing discs that will contribute to form the adult airways. Since a small subset of cells in cluster Trachea_1 expressed *btl* (Figure EV1H), which is a marker for the precursors of the adult tracheal air sacs, and is localized in the air sac primordium (ASP) (Sato & Kornberg, 2002), thus Trachea_1 could be ASP cells. Trachea_2 likely represents cells that form the transverse connective that is bound to the wing discs, and more specifically the tracheoblast cells of the spiracular branches because it expressed *ct* (Figure 1G and white arrowhead in Figure 1H) (Pitsouli & Perrimon, 2010). This information was added to the text (page 8).

*- Figures 2 and S2 should be supported by staining for gene expression, especially for new marker genes. I find the discussion about cell cluster location intriguing, but would be more convinced if supported by hard evidence to test their predictions. For example, Expression of *dpp* in Epi_16 is argued to mean that these cells belong in the anterior hinge. However, one might note that while the cells that express it do so highly, only ~50ish percent of the cells in Epi_16 are found to express it. This would be more easily explained by drop-outs if the expression levels were low.*

Response: The imaginal wing discs of mature larvae consist of group of cells showing the presumptive features of the adult pattern. Our dataset broadly supports the known biology of this primodium.

We have now included the localization of both known and new markers by immunofluorescence to strengthen the data. The expression pattern of *dpp*, *nubbin*, *wingless*, *Dad*, *svp*, and *hairy* were added in Figure 2F (pages 9-11). More importantly, we demonstrated that the localization of the new marker *Svp* fits well with predicted map of cell clusters over the disc, as it matches with the expression pattern of *Hairy* in Epi_13.

We agreed with the reviewer about the limitations of the technique and the drop-outs. We have added a note on page 13.

- Is there any way to provide higher quality expression images? I am not too familiar with the challenges of staining a wing imaginal disk, but it would seem possible to better support the spatial predictions in Figure 3 (a major result) with better imaging. Case in point: aside from image sizes being all over the place, one antibody that bothers me in particular is Ct. The authors use it to identify the DFM population, but the images do not let the reader make that distinction clearly. Some may be due to background, more may be due to low level cut expression in the IFM populations (supported by 3B). But I cannot help but wonder if RBNA-ISH might do a better job here. Furthermore, their detection in single cells is RNA, so ISH would better represent the data they describe. Also, in some panels I see the nuclear signal I expect from cut, but not in others.

Response: The work by (Sudarsan *et al*, 2001) showed that the IFM myoblasts express Vg and low levels of Ct, and the DFM myoblasts show no Vg and high levels of Ct. Since Vg antibodies were not available, we used anti-Ct antibodies. The staining is quite finicky because it shows differential expression between these two set of myoblasts (Figure 3). In order to clearly distinguish the DFM myoblasts from the IFM myoblasts, a counterstain with another marker is absolutely needed. We found that *zfh1* also shows differential expression between these two types of myoblasts (Figure 3B and 3C). *zfh1* expression is high in IFM and low in DFM. So, when we use both antibodies, anti-Ct (red) and anti-Zfh1 (cyan), we can discriminate these two set of myoblasts since DFM myoblasts will be bright red (Figure 3C, 3E, 3G, 3H, and 3I). In cases where we examined the localization of another marker using antibodies, we could not stain with both anti-Ct and anti-Zfh1 antibodies due to species cross reactivity. So, many stainings only show one marker, either anti-Ct or anti-Zfh1. And here the distinction is not straightforward. This further underscores an urgent need in the field to identify better markers for these two types of myoblasts.

A dashed yellow line was drawn on the images in Figure 3 to outline the DFM myoblast area to help the reader in interpretation of the results.

One of the advantages of working with *Drosophila* is the large collection of genetic tools available. Over the last few decades, GFP-trap, enhancer-trap, GFP-tagged tools were developed to facilitate the study of patterns of gene expression in development (Morin *et al*, 2001; Jenett *et al*, 2012; Nagarkar-Jaiswal *et al*, 2015; Kudron *et al*, 2018; Venken *et al*, 2011). Whenever possible we used antibodies to study the localization of markers over the discs (Appendix Table S4), otherwise we took advantage of these genetic tools (Appendix Table S3). If available we used two strains to validate the expression of the markers, such as *edl*, *vkg* and *Cg25C* (Figure 5 and Appendix Figure S2, pages 19-20).

In our experience the resolution of FISH in *Drosophila* tissue is not as good. However, we understand this point and we have included FISH data for *Amalgam* mRNA expression (Figure 3J-K, page 15). The localization of *Ama* probe overlaps with the *Ama-GAL4* reporter used in this study (Figure 3K, page 15).

- another problem that may just be an imaging problem is that I cannot see in Fig3H what they say I should see: "GFP was broadly expressed in both IFM and DFM precursors, thus indicating that *Ama* was expressed in IFM precursors earlier in development (Figure 3H)". First, In the downloaded images I cannot see GFP broadly in the IFM. Second, Given the strength of the GFP lineage trace in adults in 3I, I am not convinced that there is real GFP signal in the IFM at 16h APF.

Response: We have included new images in the revised manuscript (Figure 3I-K, page 15). *Ama>GFP* better illustrates *Ama* expression in DFM myoblasts and in a small subset of IFM myoblasts (yellow arrowhead, Figure 3I). This is supported by *Ama* mRNA *in situ* hybridization (Figure 3 J-K). The G-TRACE tool allowed us to trace the lineage of *Ama*-positive cells. We reported GFP expression in some IFM cells in wing discs (Figure 3K and Figure EV3C), thus indicating that *Ama* was indeed expressed in very few IFM myoblasts earlier in development. Accordingly, statements were changed in the revised text (page 15).

Images of IFM at 16 h APF were very conflicted and therefore removed from manuscript. Text was adjusted accordingly (page 15).

- no statistical tests are indicated for the screen results (Figure 5). More generally, I think the screening section should be described better. It is not clear to me from the data presented which lines were tested not only with the *Mef2* driver, but also with the *1151-Gal4* driver. Furthermore, no results for the *1151* driver are presented in the figure as far as I can see. Statements such as "the few *Mef2*>*stg*-RNAi animals that survived to adulthood were flightless" seem to go against Figure 5A that shows 50% make it to adults, and 5B shows that just about 75% land elsewhere than the bottom of the tube. Similarly, calling the RNAi animals for the elongation factor 'flightless' seems like an overstatement.

Response: We thank the reviewer for the feedback. We have reorganized data visualization and revised the text to improve the description of the screen and changed the statements (Figure 6 A-D, pages 21-22). We showed side by side the viability screen and flight test, so it is clear what RNAi lines were used for each assay. Additionally, we have extended the number of markers tested in the screen and included new data in Figure 6A-D, Appendix Table S1 and Appendix Table S2. Also, we grouped the large list of genes based on markers for each cluster. Interquartile range was added in viability plots as an indication of variability. However, in the case of the flight test, the data from different repeats were pooled together to make the assay more powerful. Since we could not include the error bars in this assay, and added the total number of animals used to plot data next to the bars, as previously done in other flight test experiment (Spletter *et al*, 2015). Appendix Table S1 and Appendix Table S2 summarize the data from the screen. Moreover, we added raw data for both assays in Source Data. Statistical tests were not added since this is mostly a qualitative screen and only severe phenotypes are relevant as previously done in other screens (Schnorrer *et al*, 2010).

- the authors propose due to the presence of *E(Spl)* genes that 3 IFM populations are notch responsive. I think this is interesting enough and should be shown by, for example, *N[ICD]* stains. This would be especially powerful if you could co-visualize it with a pan-IFM marker and maybe *Con* or *stg* versus *DNA-J1* or *Cg25c*

Response: The wing discs were stained with anti-NICD (C17.9C6, DSHB) to further study Notch response in myoblasts. Results are in concordance with previous report (Gunage *et al*, 2014). NICD was found to be localized in the membrane of the myoblasts (Figure R2A). However the use of this antibody in flies does not accurately reflect the active Notch response as it does in mammals (Zacharioudaki & Bray, 2014). Similarly, anti-NECD (458.2H, DSHB) was used to stain wing discs (Figure R2B). Unfortunately, these data were not conclusive, and therefore we did not include them in the revised manuscript. In *Drosophila*, a gold-standard read-out of Notch pathway activity are the direct Notch targets, such as the *E(spl)* genes (Zacharioudaki & Bray, 2014).

Moreover, we added more data regarding active Notch response in the revised manuscript. The localization of other *E(spl)* genes in the wing discs, such as *E(spl)m6-BFM*, *E(spl)m7-HLH* and *E(spl)mbeta-HLH*, were included in Figure 4B-C and Figure EV4F, respectively (page 17).

- The authors generally presume that their clusters have distinct spatial identities. Why? Why would clusters not represent distinct cell type identities that spatially intermingle?

Response: New data have been added to the revised manuscript to support the spatial restriction of the clusters IFM_1 and IFM_2. The localization of all three markers for IFM_1-2, including *E(spl)m3-HLH*, *E(Spl)m6-BFM* and *E(spl)m7-HLH*, showed a spatially restricted pattern as indicated by white arrowheads (Figure 4B-D, page 17). However, the expression of *LamC* and *edl*, which are markers for IFM_3, are found in cells that spatially intermingle

(Figure 5A-B). Therefore, based on the expression pattern of the markers, we conclude that some clusters have distinct spatial identities, while others are spread throughout the layers of myoblasts. We presume that the epithelial layer is contributing to this spatial restriction, as it was suggested in previous reports (Gunage *et al*, 2014; Sudarsan *et al*, 2001).

- *the argument that subpopulations may represent different differentiation states is interesting. It should be feasible to show this along pseudotime.*

Response: We have used Slingshot to examine pseudotime (Street *et al*, 2018). The IFM myoblast cells were selected for cell lineage and pseudotime inference (Figure 4E-G, page 18). Two trajectories were identified and showed the progression from muscle precursor cells to differentiated myoblasts. The findings were validated *in vivo* using the tool G-TRACE. Cells were genetically traced using markers from IFM_1-2 clusters. We confirmed that cells in IFM_1-2 clusters are in an undifferentiated state and likely represent the muscle precursor cells (AMPs). Clusters IFM_4 and IFM_5 contained more differentiated cells, and IFM_3 was in an intermediate state of differentiation. Data are included in Figure 4H-I (page 18). We hope that the reviewer would appreciate the power of combining single cell genomics with genetic, spatial and temporal lineage tracing in developing tissue.

- *I am intrigued by the fact that Ama downregulation depletes all AMP populations (DFM and IFM). This supports the TRACE experiments I criticized earlier, but more discussion is warranted.*

Response: We agree with the Reviewer that the severe reduction of IFM myoblasts in *Ama-RNAi* presents a conundrum given that *Ama* is expressed primarily in DFM myoblasts and in very few IFM myoblasts. There might be a couple of possible explanations. The scRNA-seq data revealed that, IFM myoblasts that expressed *Ama* are likely of IFM_1 cluster (Figure 3B). This result was confirmed by FISH and *Ama* reporter (Figure 3I-K, pages 15, 18). This is important because the lineage tracing experiments (Figure 4H-I, page 18) unequivocally showed that the IFM_1-2 clusters were the most undifferentiated IFM clusters. Thus, if the depletion of *Ama* affects the expansion of the precursor cells then this in turn will impact the more differentiated cell clusters IFM_4 and IFM_5, even though *Ama* is no longer expressed in these cells. Alternatively, *Ama* may impact IFM myoblasts in a non-cell autonomous manner since it is known to be secreted (Fremion *et al*, 2000). At present, we are unable to definitively prove either of these scenarios.

Minor points:

- *I do not understand what figure 1F does by itself. It should be part of 1E as visual support for the identity of the major separation, but it is not a novel result that deserves its own panel.*

Response: We think it is a good illustration of the tissue to show how myoblasts are positioned over the layer of epithelial cells. We consider it might be useful for the general reader. We hope the reviewer agrees.

- *The authors use GFP reporter lines to confirm their expression prediction (e.g. SPARC) - it seems to be work and it is believable, though RNA-ISH would have been more direct and a better readout. was there a reason for this choice?*

Response: Unfortunately, *in situ hybridization* in *Drosophila* tissue does not provide the same level of resolution as immunofluorescence. We agree that RNA-ISH is a better readout. We included FISH data in Figure 3 J-K, which overlapped nicely with the reporter of *Ama* expression (Figure 3K, page 15). Additionally, *Drosophila* model system provides an immense resource of transgenic flies to study pattern of gene expression in development (Morin *et al*, 2001; Jenett *et al*, 2012; Nagarkar-Jaiswal *et al*, 2015; Kudron *et al*, 2018; Venken *et al*, 2011). Thus, it is quite common in the *Drosophila* field to use these collections of transgenic flies that accurately reflect gene expression.

- *when indicating average expression in dot plot color, can you give units? I assume this is some highly normalized value, taking into account filtered UMI proportion per cell and per-cell UMI depth, but how*

can you get negative values? (I also assume this is average expression in cells where the gene was detected, not all cells in the cluster (as this is the Seurat default))

Response: Dot plots were generated as in the Seurat default. The expression value of each gene for each cell is normalized by the total expression and scaled, as indicated in Seurat platform. The expression of each gene is shifted, so that the mean expression across cells is 0, and is scaled, so that the variance across cells is 1. In summary, cells with a value < 0 represent cells with expression below the population mean (a value of 1 would represent cells with expression 1SD away from the population mean). This information was added to the revised text in Materials & Methods (page 37) and in legend to Figure 1 (pages 56-57).

- I believe the legend in 3A obscures data

Response: We appreciate the comment. We removed the legend and placed the labels on the side.

- I have not seen this flight test before.... how do you prevent that flies get stuck on the paper while you transfer flies, or while they fly up? in other words, why is their place of demise a landing event?

Response: This is a gold standard test that has been widely used for decades in the field. Flies are flipped into an oil-coated cylinder. The landing height is an indicator of flight performance. We measure how far flies will fall towards the bottom of the cylinder before they land on the wall of the cylinder.

Here is a short video to briefly illustrate how it is done. <https://www.jove.com/video/51223/an-improved-method-for-accurate-rapid-measurement-flight-performance>.

General notes on style and length:

LENGTH:

- The paper is written largely clearly, but would benefit from writing in a much more concise manner. For example, information such as AFPs being precursors to DFMs and IFMs is given multiple times (abstract, intro, figure legend...) and instances like that could be eliminated. Especially the figure legends should be shortened.

Response: Text was revised to make it more concise. Legends were shortened to comply with journal policy. However, we had to balance it with the policy of the journal to provide all required information in the legends in order to understand the figure.

- The current section on spatial assignments on epidermal populations - while interesting - is repetitive and hard to read. For the general reader, this section might be shortened and supported by a table. And for the expert, a supplementary note with all the details might be supplied.

Response: We consider the dataset of high value for the Drosophila community, and that's why we would like to keep it in the main manuscript. Additionally, we validated the spatial position of epithelial clusters by immunofluorescence (Figure 2F). New data are included in revised manuscript as requested by the reviewer (pages 9-11).

- There are several grammatical problems (articles, etc.) that should be addressed, but nothing that the type editors won't catch.

Response: The revised manuscript was proofread by an English native speaker. We apologize for any mistake left.

- "access" versus "assess"

Response: We thank the reviewer for catching the typo.

- To frame this investigation in terms of uncovering if ct and vg are the only two differentially expressed genes seems unreasonable. I doubt anybody would have put money on that, including he authors. Furthermore, per the author's description, it is only vestigial that would be differentially expressed between the populations... cut (they say) is expressed in both.

Response: In the revised version we emphasize that the main question of the study is to identify the main transcriptional differences between these two types of myoblast (page 4).

We revised the text and restated that *vg* is a specific marker for IFM myoblasts, and *ct* is differentially expressed between DFM and IFM myoblasts- as *ct* is high in DFM and low in IFM myoblasts (pages 4, 13).

References:

- Bageritz J, Willnow P, Valentini E, Leible S, Boutros M & Teلمان AA (2019) Gene expression atlas of a developing tissue by single cell expression correlation analysis. *Nat. Methods* **16**: 750–756
- Blochlinger K, Bodmer R, Jan LY & Jan YN (1990) Patterns of expression of Cut, a protein required for external sensory organ development in wild-type and cut mutant *Drosophila* embryos. *Genes Dev.* **4**: 1322–1331
- Evans CJ, Olson JM, Ngo KT, Kim E, Lee NE, Kuoy E, Patananan AN, Sitz D, Tran PT, Do MT, Yackle K, Cespedes A, Hartenstein V, Call GB & Banerjee U (2009) G-TRACE: Rapid Gal4-based cell lineage analysis in *Drosophila*. *Nat. Methods* **6**: 603–605
- Fremion F, Darboux I, Diano M, Hipeau-Jacquotte R, Seeger MA & Piovant M (2000) Amalgam is a ligand for the transmembrane receptor neurotactin and is required for neurotactin-mediated cell adhesion and axon fasciculation in *Drosophila*. *EMBO J.* **19**: 4463–4472
- Gunage RD, Reichert H & VijayRaghavan K (2014) Identification of a new stem cell population that generates *Drosophila* flight muscles. *Elife* **3**: 1–25
- Jenett A, Rubin GM, Ngo TTB, Shepherd D, Murphy C, Dionne H, Pfeiffer BD, Cavallaro A, Hall D, Jeter J, Iyer N, Fetter D, Hausenfluck JH, Peng H, Trautman ET, Svirskaas RR, Myers EW, Iwinski ZR, Aso Y, DePasquale GM, et al (2012) A GAL4-Driver Line Resource for *Drosophila* Neurobiology. *Cell Rep.* **2**: 991–1001
- Kudron MM, Victorsen A, Gevirtzman L, Hillier LW, Fisher WW, Vafeados D, Kirkey M, Hammonds AS, Gersch J, Ammouri H, Wall ML, Moran J, Steffen D, Szynekarek M, Seabrook-Sturgis S, Jameel N, Kadaba M, Patton J, Terrell R, Corson M, et al (2018) The modern resource: genome-wide binding profiles for hundreds of *Drosophila* and *Caenorhabditis elegans* transcription factors. *Genetics* **208**: 937–949
- Morin X, Daneman R, Zavortink M & Chia W (2001) A protein trap strategy to detect GFP-tagged proteins expressed from their endogenous loci in *Drosophila*. *Proc. Natl. Acad. Sci. U. S. A.* **98**: 15050–15055
- Nagarkar-Jaiswal S, Lee PT, Campbell ME, Chen K, Anguiano-Zarate S, Gutierrez MC, Busby T, Lin WW, He Y, Schulze KL, Booth BW, Evans-Holm M, Venken KJT, Levis RW, Spradling AC, Hoskins RA & Bellen HJ (2015) A library of MiMICs allows tagging of genes and reversible, spatial and temporal knockdown of proteins in *Drosophila*. *Elife* **2015**: 1–28
- Pitsouli C & Perrimon N (2010) Embryonic multipotent progenitors remodel the *Drosophila* airways during metamorphosis. *Development* **137**: 3615–3624
- Sato M & Kornberg TB (2002) FGF is an essential mitogen and chemoattractant for the air sacs of the *Drosophila* tracheal system. *Dev. Cell* **3**: 195–207
- Schnorrer F, Schönbauer C, Langer CCH, Dietzl G, Novatchkova M, Schernhuber K, Fellner M, Azaryan A, Radolf M, Stark A, Keleman K & Dickson BJ (2010) Systematic genetic analysis of muscle morphogenesis and function in *Drosophila*. *Nature* **464**: 287–91
- Seeger MA, Haffley L & Kaufman TC (1988) Characterization of amalgam: A member of the immunoglobulin superfamily from *Drosophila*. *Cell* **55**: 589–600
- Spletter ML, Barz C, Yeroslaviz A, Schönbauer C, Ferreira IRS, Sarov M, Gerlach D, Stark A, Habermann BH & Schnorrer F (2015) The RNA-binding protein Arrest (Bruno) regulates alternative splicing to enable myofibril maturation in *Drosophila* flight muscle. *EMBO Rep.* **16**: 178–192
- Street K, Risso D, Fletcher RB, Das D, Ngai J, Yosef N, Purdom E & Dudoit S (2018) Slingshot: Cell

- lineage and pseudotime inference for single-cell transcriptomics. *BMC Genomics* **19**: 1–16
- Stuart T, Butler A, Hoffman P, Hafemeister C, Papalexi E, Mauck WM, Hao Y, Stoeckius M, Smibert P & Satija R (2019) Comprehensive Integration of Single-Cell Data. *Cell* **177**: 1888-1902.e21
- Sudarsan V, Anant S, Guptan P, Vijayraghavan K & Skaer H (2001) Myoblast Diversification and Ectodermal Signaling in *Drosophila*. *Dev. Cell* **1**: 829–839
- Venken KJT, Schulze KL, Haelterman NA, Pan H, He Y, Evans-Holm M, Carlson JW, Levis RW, Spradling AC, Hoskins RA & Bellen HJ (2011) MiMIC: A highly versatile transposon insertion resource for engineering *Drosophila melanogaster* genes. *Nat. Methods* **8**: 737–747
- Williams JA, Bell JB & Carroll SB (1991) Control of *Drosophila* wing and haltere development by the nuclear vestigial gene product. *Genes Dev.* **5**: 2481–2495
- Zacharioudaki E & Bray SJ (2014) Tools and methods for studying Notch signaling in *Drosophila melanogaster*. *Methods* **68**: 173–182

Dear Maxim,

Since Martina is currently away from the office, I have stepped in as the secondary editor of your manuscript.

Thank you for submitting your revised manuscript. It has now been seen by two of the original referees.

As you can see, the referees find that the study is significantly improved during revision and recommend publication. Before I can accept the manuscript, I need you to address some minor points below:

- Please address the remaining minor concern of referee #2.
- Our data editors note that figure legends of Figure 6A and C state $n=2$ independent experiments, but error bars and p-values are calculated. Normally, statistics should not be calculated if $n<3$. Ideally, the averages of 3 independent experiments should be the basis for statistics, and the error bars should be calculated based on the variance of these averages. If only 2 experiments were performed, we do offer authors to show all data points in the graphs along with their mean, but no error bars and no p-values, although it would be better to show the average of all 3 experiments with error bars and p-values. Please let me know how you would like to proceed.
- Please upload the source data as one file per figure.
- The source data labelled as Figs 4E, F and 7C don't seem to match the panels. Please check.
- Please make the Dropseq scRNAseq data (GSE138626) publicly accessible.
- All articles published beginning 1 July 2020, the EMBO Reports reference style changed to the Harvard style for all article types. Details and examples are provided at <https://www.embopress.org/page/journal/14693178/authorguide#referencesformat> Please update the reference style accordingly.
- Please complete the funder information in the manuscript submission system.
- Movies need to be ZIPped with their legends. The legends should be removed from the Article file.
- We noted that scale bars of Figure 1D and G are either missing or not visible enough.
- Papers published in EMBO Reports include a 'Synopsis' to further enhance discoverability. Synopses are displayed on the html version of the paper and are freely accessible to all readers. The synopsis includes a short standfirst summarizing the study in 1 or 2 sentences that summarize the key findings of the paper and are provided by the authors and streamlined by the handling editor. I would therefore ask you to include your synopsis blurb.
- Thank you for providing an image for the synopsis. I note that it is currently a bit too crowded with text and many images and it will not look clear when we resize it to 550x400 pixels (as per technical requirements). Please simplify the image.
- Our production/data editors have asked you to clarify several points in the figure legends (see attached document). Please incorporate these changes in the attached word document and return it with track changes activated.

Thank you again for giving us to consider your manuscript for EMBO Reports, I look forward to your minor revision.

Kind regards,

Deniz

--

Deniz Senyilmaz Tiebe, PhD
Editor
EMBO Reports

Referee #2:

As detailed in my initial review, this is a very valuable manuscript cataloguing the transcriptional diversity of the many *Drosophila* larval myoblasts present in wing discs, which will form the various thoracic adult muscle fibers. It also classifies all the wing disc epithelial cells as a bonus. Hence, the paper generated a large resource, the myoblast cell atlas, which identified distinct myoblast sub-populations that enabled to delineate developmental pseudo-time of IFM precursor development. It identified various interesting potential regulators of myoblast development, some of which have been functionally verified; one novel gene called *Ama* in more detail. Thus, this manuscript will inspire various follow up works of the community.

The revised manuscript clarified most of the confusion about nomenclature of the clusters and the incorporation of the classical myoblast markers into Figure 1G helped significantly. The presentation and verification of the various clusters is now very convincing and ready for publication. I still wonder why the discontinuous numbering of the epithelial clusters was kept in the revision.

The flight muscle selector gene *Salm* is indeed only expressed in myotubes and not yet in myoblasts (Schönbauer et al. 2010), fitting to the authors findings.

The newly added genetic data do corroborate a function for *Ama* in muscle development. Specifically, *Ama* functions during maintenance or proliferation of DFM and IFM forming myoblasts during larval stages. *Ama* is likely also affecting other muscle classes in the legs and abdomen thus explaining the lethality found after knock-down (IFMs and DFMs are not required for viability). It would have been nice to see a better quantification of the genetic rescue of the *Ama* RNAi induced lethality when re-expressing *Ama* cDNA. Are the myoblast numbers in L3 wing discs rescued, too? Are adult flight muscles recovered? These results would further corroborate a function of *Ama* in muscles, as said the lethality won't be caused by IFM or DFM phenotypes.

Referee #3:

The revised manuscript by Zappia et al. "A cell atlas of adult muscle precursors uncovers early events in fiber-type divergence in *Drosophila*" has much improved. After reading both the extensive comments to the reviewers and the revised version of the manuscript, I would like to congratulate the authors on a job well done.

The paper is now clearer than it was and I believe that the concerns by all reviewers were sufficiently addressed; indeed, most were very convincingly addressed.

Especially the stainings have improved, the additional markers are convincing, and the presentation and interpretation of the screen has very much improved. I appreciate the added pseudo-temporal analysis, and while it may have turned out less impactful than I would have guessed, it is an

important piece of information to put the relevant clusters and cells in both spatial, and now also temporal context ... to a degree. Furthermore, the added note about spatial separation vs intermingling of differentiated later cell types is well-warranted.

Taken together, the manuscript presents an important resource and combines this with interesting biological insights into the developmental trajectory of flight muscles.

I recommend publication without further review.

Referee #2:

As detailed in my initial review, this is a very valuable manuscript cataloguing the transcriptional diversity of the many *Drosophila* larval myoblasts present in wing discs, which will form the various thoracic adult muscle fibers. It also classifies all the wing disc epithelial cells as a bonus. Hence, the paper generated a large resource, the myoblast cell atlas, which identified distinct myoblast sub-populations that enabled to delineate developmental pseudo-time of IFM precursor development. It identified various interesting potential regulators of myoblast development, some of which have been functionally verified; one novel gene called *Ama* in more detail. Thus, this manuscript will inspire various follow up works of the community.

The revised manuscript clarified most of the confusion about nomenclature of the clusters and the incorporation of the classical myoblast markers into Figure 1G helped significantly. The presentation and verification of the various clusters is now very convincing and ready for publication. I still wonder why the discontinuous numbering of the epithelial clusters was kept in the revision.

The flight muscle selector gene *Salm* is indeed only expressed in myotubes and not yet in myoblasts (Schönbauer et al. 2010), fitting to the authors findings.

The newly added genetic data do corroborate a function for *Ama* in muscle development. Specifically, *Ama* functions during maintenance or proliferation of DFM and IFM forming myoblasts during larval stages. *Ama* is likely also affecting other muscle classes in the legs and abdomen thus explaining the lethality found after knock-down (IFMs and DFMs are not required for viability).

It would have been nice to see a better quantification of the genetic rescue of the *Ama* RNAi induced lethality when re-expressing *Ama* cDNA. Are the myoblast numbers in L3 wing discs rescued, too? Are adult flight muscles recovered? These results would further corroborate a function of *Ama* in muscles, as said the lethality won't be caused by IFM or DFM phenotypes.

Response: We thank the reviewer for his/her comments. We are pleased that the reviewer noted the value of our study to the scientific community.

Figure R3: Confocal single plane images of leg imaginal discs in *Ama*[NP1297]>GFP third instar larva stained with the myoblast marker *zfh1* (red) and DAPI.

The reference to Schönbauer et al. 2011 was added to the final version of the manuscript on page 14.

The referee asked whether we performed further studies on *Ama* rescue, such as analyzing the number of myoblasts in L3 wing discs and the structure of the adult flight muscles. While this would be informative, such quantification is not straightforward because the number of fully recovered animals was relatively low (Figure 7A). Therefore, we did not pursue this analysis any further. We hope that the reviewer agrees.

As referee mentioned, *Ama* is most likely affecting other muscles as well. Indeed, the expression of *Ama*>GFP was detected in myoblasts of L3 leg imaginal discs (Figure R3). However, we do not know yet in what tissues *Ama* is required for viability.

Referee #3:

The revised manuscript by Zappia et al. "A cell atlas of adult muscle precursors uncovers early events in fiber- type divergence in *Drosophila*" has much improved. After reading both the extensive comments to the reviewers and the revised version of the manuscript, I would like to congratulate the authors on a job well done.

The paper is now clearer than it was and I believe that the concerns by all reviewers were sufficiently addressed; indeed, most were very convincingly addressed.

Especially the stainings have improved, the additional markers are convincing, and the presentation and interpretation of the screen has very much improved. I appreciate the added pseudo-temporal analysis, and while it may have turned out less impactful than I would have guessed, it is an important piece of information to put the relevant clusters and cells in both spatial, and now also temporal context to a degree. Furthermore, the added note about spatial separation vs intermingling of differentiated later cell types is well-warranted.

Taken together, the manuscript presents an important resource and combines this with interesting biological insights into the developmental trajectory of flight muscles.

I recommend publication without further review.

Response: We thank the referee for the positive response and recommending publication without further review.

Maxim Frolov
University Illinois at Chicago
Biochemistry and Molecular Genetics
900 S Ashland Ave, MBRB2352
Chicago, IL 60607
United States

Dear Maxim,

Sorry for the slight delay in handling your further revised manuscript - I just came back to the office. But I have now gone through all minor changes that were required and am now very pleased to accept your manuscript for publication in the next available issue of EMBO reports. Thank you for your contribution to our journal.

At the end of this email I include important information about how to proceed. Please ensure that you take the time to read the information and complete and return the necessary forms to allow us to publish your manuscript as quickly as possible.

As part of the EMBO publication's Transparent Editorial Process, EMBO reports publishes online a Review Process File to accompany accepted manuscripts. As you are aware, this File will be published in conjunction with your paper and will include the referee reports, your point-by-point response and all pertinent correspondence relating to the manuscript.

If you do NOT want this File to be published, please inform the editorial office within 2 days, if you have not done so already, otherwise the File will be published by default [contact: emboreports@embo.org]. If you do opt out, the Review Process File link will point to the following statement: "No Review Process File is available with this article, as the authors have chosen not to make the review process public in this case."

Should you be planning a Press Release on your article, please get in contact with emboreports@wiley.com as early as possible, in order to coordinate publication and release dates.

Thank you again for your contribution to EMBO reports and congratulations on a successful publication. Please consider us again in the future for your most exciting work.

Kind regards,
Martina

Martina Rembold, PhD
Editor
EMBO reports

THINGS TO DO NOW:

You will receive proofs by e-mail approximately 2-3 weeks after all relevant files have been sent to our Production Office; you should return your corrections within 2 days of receiving the proofs.

Please inform us if there is likely to be any difficulty in reaching you at the above address at that time. Failure to meet our deadlines may result in a delay of publication, or publication without your corrections.

All further communications concerning your paper should quote reference number EMBOR-2019-49555V3 and be addressed to emboreports@wiley.com.

Should you be planning a Press Release on your article, please get in contact with emboreports@wiley.com as early as possible, in order to coordinate publication and release dates.

Corresponding Author Name: Maxim Frolov

Manuscript Number: EMBOR-2019-49555-T